# Can Large Language Models Capture Dissenting Human Voices?

**Noah Lee**[*]
KAIST AI
noah.lee@kaist.ac.kr

**Na Min An**[*]
KAIST AI
naminan@kaist.ac.kr

**James Thorne**
KAIST AI
thorne@kaist.ac.kr

## Abstract

Large language models (LLMs) have shown impressive achievements in solving a broad range of tasks. Augmented by instruction fine-tuning, LLMs have also been shown to generalize in zero-shot settings as well. However, whether LLMs closely align with the human disagreement distribution has not been well-studied, especially within the scope of natural language inference (NLI). In this paper, we evaluate the performance and alignment of LLM distribution with humans using two different techniques to estimate the multinomial distribution: Monte Carlo Estimation (MCE) and Log Probability Estimation (LPE). As a result, we show LLMs exhibit limited ability in solving NLI tasks and simultaneously fail to capture human disagreement distribution. The inference and human alignment performances plunge even further on data samples with high human disagreement levels, raising concerns about their natural language understanding (NLU) ability and their representativeness to a larger human population.[1]

## 1 Introduction

Natural language inference (NLI) has long served as a fundamental testbed to evaluate the ability of a model to recognize entailment and capture plausible inference relations between pairs of sentences (Dagan et al., 2006; Bowman et al., 2015; Williams et al., 2018). When constructing datasets, conventional processes result in a single label per instance even if multiple annotators contribute, which limits the full representation of diverse opinions that might arise in a larger human population. Thus, recent datasets have become more attentive to incorporating multiple interpretations (Pavlick and Kwiatkowski, 2019; Nie et al., 2020b; Glockner et al., 2023) to capture dissenting human opinions.

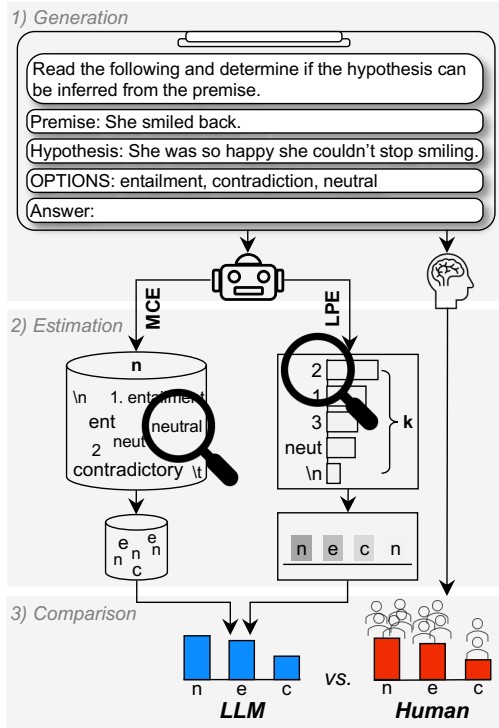

Figure 1: The Proposed LLM Distribution Estimation Techniques, MCE and LPE. We estimate LLM disagreement with either MCE or LPE utilizing generated LLM outputs and compare the estimated LLM distribution with human disagreement distribution.

Meanwhile, instruction fine-tuning large language models (LLMs) has elicited remarkable generalizability to diverse unseen tasks (Zhao et al., 2023). Not only can they generate free-form texts, but they can also select one answer from multiple options given in the input prompt. However, while many works study user interaction and conversational usage (Liang et al., 2022), limited works evaluate these instruction-following LLMs on a foundational NLI task. Therefore, we aim to answer the following questions: Can LLMs capture dissenting voices that naturally arise in the dataset? Are LLMs representative of the voices of the annotators in inference tasks?

---

[*]Equal contribution

[1]The source code for the experiments is available at https://github.com/xfactlab/emnlp2023-LLM-Disagreement.

With this in mind, we jointly assess on a number of instruction-following LLMs, Flan-T5 (Chung et al., 2022), Flan-UL2 (Tay et al., 2022), OPT-IML-Max (Iyer et al., 2022), and GPT-3 (Ouyang et al., 2022), on their performance on human opinion distribution datasets - ChaosNLI (Nie et al., 2020b) and PK2019 (Pavlick and Kwiatkowski, 2019). For the process of using the model output distribution as an estimate of human disagreement distribution, we offer novel estimation methods: Monte Carlo Estimation (MCE) and Log Probability Estimation (LPE) (Figure 1).

We find that the state-of-the-art GPT-3 model does not outperform smaller models such as fine-tuned BERT (Devlin et al., 2019) and partially fine-tuned Flan-T5-XXL in solving inference problems. Furthermore, it yields higher Jensen-Shannon Distance (JSD) (Endres and Schindelin, 2003) and Distribution Calibration Error (DCE) (Baan et al., 2022) than BERT for the ChaosNLI datasets. Each model is optimized using different estimation methods and prompt types, where GPT/Flan-T5-XXL attains the best performances in NLI capability and human alignment when using LPE/MCE. Our paper's contributions are as follows:

- To the best of our knowledge, we are the first to test generative LLMs jointly on the performance and human disagreement on NLI.

- We suggest two probability distribution estimation techniques for LLMs to represent disagreement and perform empirical evaluations to with respect to the human disagreement distribution.

- We study the model sensitivity to estimation methods and prompt types to demonstrate how these contribute to the ability of models to represent human-level disagreement for NLI.

## 2   Related Works

### 2.1   Disagreement in NLI

Considering only a single label in NLI datasets is bound to fail in capturing the diverse range of user opinions and could lead to misrepresentations of language models. To measure inherent human disagreements in NLI, Nie et al., 2020b and Pavlick and Kwiatkowski, 2019 collected large number of human annotations (*e.g.*, 100 and 50 annotations for ChaosNLI and PK2019) per instance for common NLI datasets such as SNLI (Bowman et al., 2015) and MultiNLI (Williams et al., 2018). When taking the majority vote from these additional annotations, 22% of the instances exhibited a change in label compared to the original dataset (Nie et al., 2020b).

To characterize and reproduce the extent of human disagreement in NLI tasks, previous works directly fine-tuned language models (Nie et al., 2020b) and implemented distribution estimation methods (Zhou et al., 2022) using the labeled data. Other studies have constructed losses to better calibrate the ambiguity (Meissner et al., 2021) and proposed an ensemble of models to detect disagreeing samples (Zhang and de Marneffe, 2021).

For measuring the distance between two distributions, Kullback–Leibler (KL) Divergence (Kullback and Leibler, 1951) or its symmetric version, Jensen-Shannon Distance (JSD) (Endres and Schindelin, 2003) are widely used. Baan et al., 2022 argued that Expected Calibration Error (ECE), the difference between the average accuracy and confidence (Naeini et al., 2015; Guo et al., 2017), cannot capture inherent human disagreement. Therefore, for models to better calibrate to human disagreement, accuracy-agnostic metrics such as DCE have been introduced (Baan et al., 2022).

### 2.2   Alignment of Instruction-tuned LLMs

LLMs have demonstrated the ability to follow examples provided in-context (Brown et al., 2020) and have further been developed to follow natural language instructions (Mishra et al., 2022; Ouyang et al., 2022; Chung et al., 2022). Instruction-following LLMs are fine-tuned with various tasks and are expected to generalize well to tasks the model was not trained on (Zhao et al., 2023). For example, GPT-3 is fine-tuned using reinforcement learning with human feedback to produce responses that align with human values (Ouyang et al., 2022). Despite such efforts, Santurkar et al., 2023 identified that LLMs capture only a single perspective, exhibiting left-leaning tendencies and excluded demographic groups. Here, we study whether LLMs appropriately reflect a diversity of viewpoints in the NLI task setting.

## 3   Methods

We estimate and quantify dissenting human voices using the multinomial soft-label distribution of LLMs with two proposed methods:

## 3.1 Log Probability Estimation (LPE)

We use a single instance returning log probabilities of top-k[2] token candidates to estimate the categorical distribution of the labels. This method sums over all valid options[3] ($v_j$) to estimate the model probability for class j, a method often adopted in a multiple-choice style evaluation of generative language models (Hendrycks et al., 2021; Santurkar et al., 2023). Although the LPE method requires a single generation for each instance, it cannot be applied to all types of models[4]. Additionally, the method is limited in cases where more than one token is generated as it requires exhaustive mapping of the determining token probability. Furthermore, as models only return probabilities for top-k tokens, there is an unknown non-constant probability mass. We estimate this as follows, where $C$ is the total number of classes of the task:

$$p(\hat{y}_j|x) \approx \frac{\sum_{i=1}^{k} \exp l p_i \cdot \mathbb{1}_{i \in v_j}}{\sum_{j=1}^{C} \sum_{i=1}^{k} \exp l p_i \cdot \mathbb{1}_{i \in v_j}} \quad (1)$$

## 3.2 Monte Carlo Estimation (MCE)

Decoding strategies such as beam search or greedy search do not exploit the full distribution of the possible generation options. Furthermore, API-based language model services limit the number of returned token-level probabilities. Alternatively, to reconstruct the distribution of outputs from generative LLMs, we introduce an intuitive way that samples a large number[5] of generated outputs considering the valid options[6] ($v_j$) for class j. This method is based on a Monte Carlo method (Metropolis and Ulam, 1949) to estimate the probability distribution. Even though the MCE method can be computationally expensive, it can be applied to any model and prompt type to capture the multinomial distribution of a classification setting. MCE is defined as follows:

$$p(\hat{y}_j|x) \approx \frac{1}{n} \sum_{i=1}^{n} \mathbb{1}_{i \in v_j} \quad (2)$$

---

[2]k is set to 5 for all the models to match the maximum logprobs size of OpenAI Completion API.

[3]See Appendix C for examples.

[4]GPT-3.5-Turbo does not support `logprobs`.

[5]Sample size of 100 is heuristically chosen to match the size of human annotation for ChaosNLI.

[6]See Appendix C for examples.

## 4 Experimental Design

### 4.1 Data

First, we test the inference ability of LLMs in challenging datasets, ANLI (Adversarial NLI) (Nie et al., 2020a) and QNLI (Wang et al., 2018). We opt for the round 3 version of ANLI (n = 1,200), which contains more contexts from diverse domains such as Wikipedia, Common Crawl, StoryCloze (Mostafazadeh et al., 2016), and CBT (Hill et al., 2016). QNLI (Wang et al., 2018) (n = 5,463) is converted from the Stanford Question Answering Dataset (SQuAD) (Rajpurkar et al., 2016) to an NLI dataset, and the task is to decide whether the sentence contains the answer to the question.

Second, we jointly evaluate LLMs on ChaosNLI (Nie et al., 2020b), and PK2019 (Pavlick and Kwiatkowski, 2019) to examine both the accuracy and how the model distribution aligns with the human disagreement distribution. These datasets consist of two task settings: ChaosNLI-$\alpha$ (n = 1,532), where models must select one of the two hypotheses, and ChaosNLI-S (n = 1,514), M (n = 1,599), and a subset[7] of PK2019 (n = 299) where models must assign the relationship (*e.g.*, entailment, contradiction, or neutral) for a pair of premise and hypothesis. We also pick out a challenging subset of the ChaosNLI datasets, which we denote as HighChaosNLI, consisting of the top 100 samples having the greatest human disagreement level.

Lastly, to trace possible causes of the disagreement occurring in LLMs, we use the round 1 version of the DisagreementNLI dataset (n = 318), where the samples from ChaosNLI are annotated with one of the 10 categories (*e.g.*, probabilistic) of potential sources of disagreement (Jiang and Marneffe, 2022). While the primary focus is slanted towards identifying why humans disagree, we utilize and link the disagreement taxonomy to uncover whether the disagreement in LLMs aligns with those of humans.

### 4.2 Models

We categorize numerous LLMs with varying levels of supervision on the NLI task[8]: Full Exposure (FE), Partial Exposure (PE), Minimum/Unknown Exposure (MUE), and No Exposure (NE). For FE models, we follow the baseline setup of Nie et al., 2020b by fine-tuning BERT (340M) (Devlin et al.,

---

[7]JOCI & DNC datasets of PK2019 are discarded as the annotation setting greatly varies from ChaosNLI.

[8]See Appendix B for more details.

| Model | LPE (NS) | | | MCE (NS) | | | MCE (OS) | | |
|---|---|---|---|---|---|---|---|---|---|
| | Acc↑ | JSD↓ | DCE↓ | Acc↑ | JSD↓ | DCE↓ | Acc↑ | JSD↓ | DCE↓ |
| **Flan-T5-L** (780M) | 59.3 | 0.293 | 0.326 | **62.3** | **0.289** | **0.321** | 59.7 | 0.290 | 0.322 |
| **Flan-T5-XL** (3B) | 65.7 | 0.253 | 0.282 | **72.0** | **0.236** | **0.254** | 70.3 | 0.238 | 0.256 |
| **Flan-T5-XXL** (11B) | 68.7 | 0.258 | 0.277 | 71.0 | 0.263 | 0.277 | **74.3** | **0.232** | **0.244** |
| **Flan-UL2** (20B) | 67.7 | 0.260 | 0.281 | 72.3 | 0.247 | 0.259 | **76.0** | **0.241** | **0.246** |
| **OPT-IML-M-S** (1.3B) | 57.0 | **0.294** | 0.337 | 54.7 | 0.312 | 0.354 | **59.3** | 0.298 | **0.337** |
| **OPT-IML-M-L** (30B) | 72.0 | 0.273 | 0.286 | 62.0 | 0.280 | 0.303 | **72.7** | **0.233** | **0.252** |
| **GPT-3-D3** (175B) | 66.7 | **0.330** | **0.345** | **67.0** | 0.334 | 0.349 | 58.0 | 0.344 | 0.376 |
| **GPT-3-D2** (175B) | **64.0** | 0.282 | 0.317 | 62.7 | **0.279** | **0.313** | 49.3 | 0.315 | 0.364 |
| **Stable Vicuna** (13B) | **45.7** | **0.328** | **0.379** | 43.7 | 0.504 | 0.568 | 41.7 | 0.502 | 0.567 |

Table 1: Human Alignment Performances of LLMs on Subsets of ChaosNLI Datasets with Different Estimation Methods - LPE/MCE (Prompt Types - Shuffled NS/OS). We present the average results of ChaosNLI-$\alpha$, S, and M; for each dataset, we randomly sample 100 instances. The model categorizations are the same as Table 2. Bold texts indicate the best value for each model and metric.

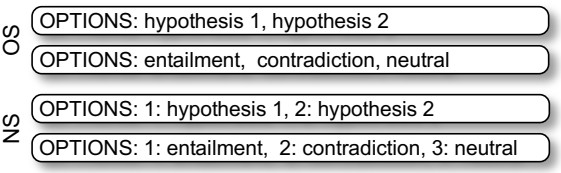

Figure 2: Two Prompt Types - OS and NS for Two Types of Tasks. The former does not have a number in front of each option choice.

2019) and RoBERTa (355M) (Liu et al., 2019). Since instruction-following LLMs do not have full supervision of NLI, we assign these LLMs to one of the PE, MUE, and NE models.

First, the PE models include Flan-T5 (780M, 3B, 11B) (Chung et al., 2022), Flan-UL2 (20B) (Tay et al., 2022), and OPT-IML-Max (1.3B and 30B)[9] (Iyer et al., 2022). We label GPT-3-D2 (text-davinci-002) and GPT-3-D3 (text-davinci-003) (175B) (Ouyang et al., 2022) as MUE models because, although the models are variants from Ouyang et al., 2022, it is unknown to which extent the model is exposed to NLI tasks. Finally, the sole NE model that we test is Stable Vicuna (13B) (Chiang et al., 2023)[10] because it has no exposure to NLI tasks. All the hyperparameters we use to generate the outputs of these models are listed in Appendix A.

## 4.3 Prompt Types

We adopt mostly the same prompt template[11] across different types of models within each sub-dataset. Within a dataset and a model, we test two types of prompts: (1) Option Selection (OS), in which the model has to predict the name of the class label for the entailment relation, and (2) Number Selection[12] (NS), in which the model has to select the number assigned to the relationship class (Figure 2). Additionally, as LLMs are known to be sensitive to minor input modifications (Liang et al., 2022; Sun et al., 2023), we test the effect of prompt variations over a single prompt.

NS requires the model to predict a single token of a target number and can be used with both MCE and LPE. OS, on the other hand, is not considered in the LPE formulation to encourage a scalable, comprehensive generation strategy to estimate human disagreement distribution since if we allow LPE-OS, the token-specific probability of a model output which may vary by instance/dataset/task has to be mapped per class. We implement random ordering of the options in the prompt, as also mentioned in Santurkar et al., 2023, to mitigate the sensitivity due to the order of the options, which we call shuffled OS and NS throughout the paper.

## 4.4 Metrics

We investigate the distribution differences between humans and LLMs at the sample level with JSD, which is a symmetric version of KL divergence

---

[9]These models will be referred as OPT-IML-M-S and OPT-IML-M-L for convenience.

[10]Results of poor-performing models such as Stanford Alpaca (7B) and Dolly-v2 (12B) are not reported.

[11]Refer to Appendix F for specific prompt examples.

[12]A multiple-choice format similar to the prompt suggested in the MMLU Benchmark (Hendrycks et al., 2021)

| Model | ANLI-R3 | QNLI | ChaosNLI-$\alpha$ | ChaosNLI-S | ChaosNLI-M | PK2019 |
|---|---|---|---|---|---|---|
| **Chance** | 33.3 | 50.0 | 50.0 | 33.3 | 33.3 | 33.3 |
| *Full Exposure (FE)* | | | | | | |
| **BERT-L**[*] (340M) | 43.5 | 92.7 | 68.2 (+0.2) | 73.8 (+1.2) | 56.9 (-4.3) | - |
| **RoBERTa-L**[*] (355M) | 44.4 | **98.9** | 83.7 (+1.6) | 78.7 (+3.8) | 63.5 (-3.9) | - |
| *Partial Exposure (PE)* | | | | | | |
| **Flan-T5-L** (780M) | 46.3 | 90.2 | 73.1 (+1.9) | 54.8 (-4.6) | 59.7 (+7.8) | 76.6 (+6.5) |
| **Flan-T5-XL** (3B) | 54.3 | 93.1 | 83.3(+1.2) | 71.2 (+1.1) | 60.2 (+1.0) | 76.9 (-7.4) |
| **Flan-T5-XXL** (11B) | **58.2** | 93.7 | 84.9 (+1.6) | 67.9 (+0.8) | **72.6** (+8.5) | **82.1** (-0.6) |
| **Flan-UL2** (20B) | 56.6 | 94.9 | **86.5** (+1.8) | **79.9** (+6.4) | 71.7 (+4.8) | 74.6 (-14.3) |
| **OPT-IML-M-S** (1.3B) | 34.6 | 80.6 | 53.6 (-0.7) | 66.1(+7.2) | 50.3 (+1.2) | 57.5 (-3.1) |
| **OPT-IML-M-L** (30B) | 38.5 | 70.4 | 72.7 (-1.8) | 77.1 (+7.3) | 65.4 (-3.5) | 68.3 (-14.5) |
| *Minimal/Unknown Exposure (MUE)* | | | | | | |
| **GPT-3-D3** (175B) | 47.8 | 79.0 | 76.5 (+2.3) | 62.7 (+5.6) | 63.3 (+9.1) | 69.5 (-0.2) |
| **GPT-3-D2** (175B) | 44.8 | 77.1 | 72.6 (+1.5) | 56.3 (+6.0) | 49.9 (-0.7) | 45.5 (-10.6) |
| *No Exposure (NE)* | | | | | | |
| **Stable Vicuna** (13B) | 33.5 | 49.5 | 55.6 (+2.1) | 34.2 (-5.6) | 45.4 (+8.5) | 61.2 (+14.5) |

Table 2: Inference Performances of LLMs on Various Datasets. We use MCE (n = 5 for ANLI-R3/QNLI and n = 500 for ChaosNLI/PK2019) with shuffled OS. For GPTs and Stable Vicuna, we use LPE with shuffled NS. The values inside the parentheses indicate the accuracy change from the old to new labels. We report the accuracy results of the FE models (∗) from Nie et al., 2020a for ANLI-R3, Devlin et al., 2019 and Liu et al., 2019 for BERT-L and RoBERTa in QNLI, and Nie et al., 2020b for ChaosNLI datasets. All the outputs are averaged over three runs, and bold and underlined texts indicate the first and the second best value for each column.

(Endres and Schindelin, 2003). In addition, we evaluate human uncertainty with DCE (Baan et al., 2022) to examine how the tendencies of these two measures compare.

$$\mathbf{JSD(p||q)} = \sqrt{\frac{\mathbf{KL(p||m) + KL(q||m)}}{2}}$$

$$\mathbf{DCE(p, q)} = \frac{1}{2}||\mathbf{p - q}||_1$$

where $\mathbf{KL(p||q)} = \sum_i p_i \log(\frac{p_i}{q_i})$, $\mathbf{m} = \frac{\mathbf{p+q}}{2}$

## 5  Results

**LLMs are sensitive to different estimation methods and prompt types.** To select the optimal estimation methods and prompt types for each model, we examine three cases[13] - (1) LPE (NS), (2) MCE (NS), and (3) MCE (OS) for 100 randomly selected examples in ChaosNLI subsets (Table 1). All the PE models perform the best using MCE (OS) or MCE (NS). Meanwhile, GPT-3-D3 performs better using LPE (NS) than either MCE method, hinting that larger models (>100B) may not need costly methods to estimate the model distribution. Similarly, for GPT-3-D2 and Stable Vicuna, a drastic

[13] We exclude LPE (OS) due to the reason outlined in Section 4.3.

negative effect of using MCE methods is exhibited, especially when using OS. Hence, we choose MCE (OS) for the PE models and LPE (NS) for the MUE and NE models.

**The NLI capability of LLMs does not only increase due to model size.** In Table 2, even though GPT-3-D3 has the largest parameters (175 billion) and surpasses GPT-3-D2 and Stable Vicuna, its accuracy is significantly outperformed by the PE models across ANLI-R3, QNLI, ChaosNLI, and PK2019 datasets. For ChaosNLI-S especially, GPT-3-D3 shows comparably lower performances than any FE and PE models. The leading PE models are Flan-UL2 and Flan-T5-XXL across most of the tested datasets (Table 2). The best PE model achieves 9 to 15% higher accuracy in ANLI-R3/ChaosNLI-M than the best FE model (*i.e.*, RoBERTa-L). However, Flan-T5-UL2 is marginally higher than RoBERTa-L by 1 to 3 % in ChaosNLI-$\alpha$/S, and Flan-T5-XXL even achieves 9.1% higher than RoBERTa-L for ChaosNLI-M. Within the Flan-T5 family, scaling the model leads to enhanced inference performances. However, the largest model across all the tested models - GPT-3-D3 does not always attain the best accuracy, suggesting that model size alone is not a critical factor for performance on NLI.

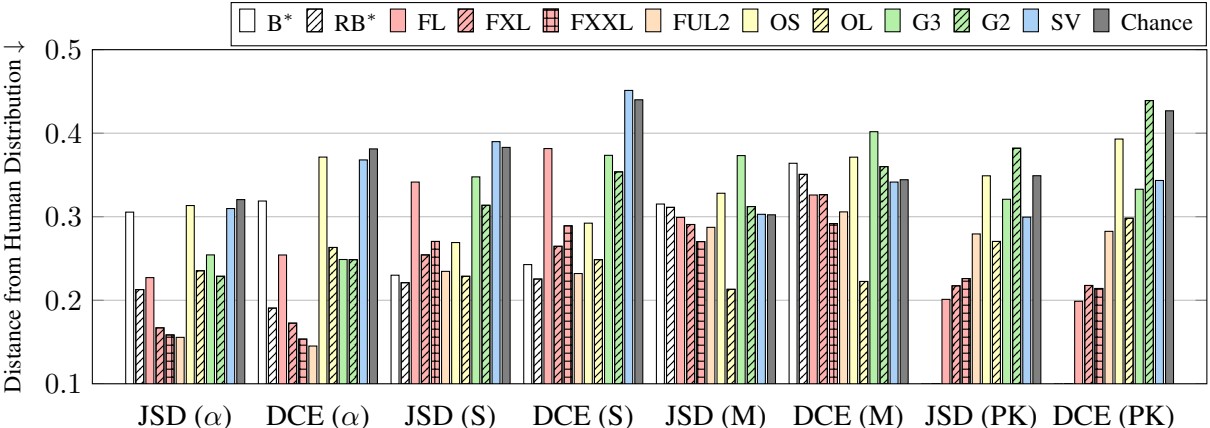

Figure 3: Human Alignment Performances of LLMs on ChaosNLI and PK2019 Datasets. The model categorizations and estimation methods are the same as Table 2. All the outputs are averaged over three runs. We additionally visualize pairwise model similarity using JSD in Appendix D.

**Does multiple annotation help?** For most of the models making inferences on the re-annotated datasets of ChaosNLI, improvements in NLI accuracy are observed, with the exception of OPT-IML-M-S. (Refer to the values inside parentheses in Table 2). This supports the necessity of having increased multiple annotations for tasks that humans are expected to disagree with. Also, it is noticeable how all these models, even if they were exposed to a sample of the train set with the original label, show better performances in the newly annotated ChaosNLI. However, we detect an accuracy decrease between the old and new labels in the PK2019 dataset for most of the models except for Flan-T5-L and Stable Vicuna. We hypothesize this is due to the way in which the final label was selected in Pavlick and Kwiatkowski, 2019: annotators were asked to select an interval score which was later manually discretized.

**Alignment with human disagreement is not always better for larger models.** To examine how closely the estimated distribution of LLMs aligns with the human disagreement distribution, we compare sample-level measures of JSD and DCE between humans and LLMs (Figure 3). Similar to the accuracy results (Table 2), GPT-3-D3 fails to align with the human label distribution compared to some well-performing PE models, such as Flan-T5-XXL and Flan-T5-UL2. Also, each model displays a similar tendency between JSD and DCE, suggesting that either one of the metrics might be enough to measure human alignment.

As can be observed in Figure 3, none of the LLMs show less JSD/DCE values than RoBERTa-

L in ChaosNLI-$\alpha$/S. Within LLMs, there is no one leading model that performs well across all datasets. For example, while Flan-UL2 scores the lowest JSD/DCE value in the ChaosNLI-$\alpha$ dataset, OPT-IML-M-L shows the lowest distance from human distribution in the ChaosNLI-M dataset. It is important to note that GPT-3-D3 shows worse JSD/DCE than RoBERTa-L for all ChaosNLI datasets, and it even performs worse than Stable Vicuna in ChaosNLI-M. Intriguingly, the Flan-T5 family benefits from scaling model size in ChaosNLI datasets, but Flan-T5-large does not show the highest JSD/DCE in PK2019 datasets.

**Effect of Human Entropy on LLM Disagreement** We filter out a challenging subset, High-ChaosNLI, which is the top 100 selected samples with the highest human disagreement levels based on the entropy of each instance. We observe a plunge in accuracy as well as a rise in JSD/DCE for every model (Table 3) compared to the human alignment performances for full datasets in Table 2. Still, the leading model concerning inference ability (*i.e.*, Flan-T5-XXL) is unchanged, obtaining the highest accuracy of 52% in HighChaosNLI. On the other hand, it is notable how Stable Vicuna displays the lowest JSD/DCE compared to the other models (Table 3). Nevertheless, with the hint of the worst accuracy out of all the models for full ChaosNLI datasets (Figure 3) and high entropy levels (Figure 4), we conclude that it is a mere coincidence that Stable Vicuna exhibits the best performance in terms of human alignment performances in High-ChaosNLI dataset (Table 3).

We further attempt to investigate the possible

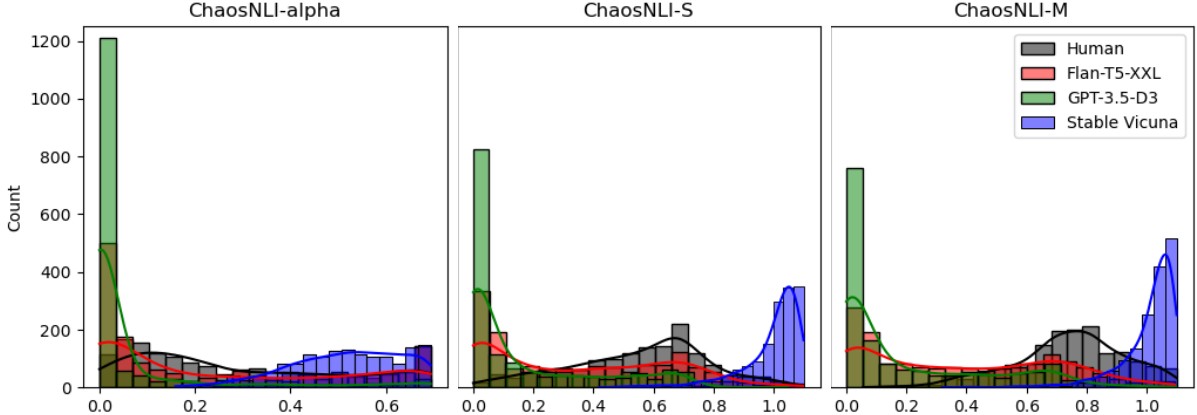

Figure 4: Histogram of Human and LLM Entropy Levels for ChaosNLI Datasets. The distributions of Flan-T5-XXL and GPT-3-D3/Stable Vicuna are estimated using MCE (OS) and LPE (NS), respectively, same as Table 2.

| Model | HighChaosNLI | | |
| --- | --- | --- | --- |
| | Acc↑ | JSD↓ | DCE↓ |
| **Flan-T5-L** (780M) | 44.0 | 0.256 | 0.318 |
| **Flan-T5-XL** (3B) | 48.0 | 0.268 | 0.336 |
| **Flan-T5-XXL** (11B) | **52.0** | 0.300 | 0.362 |
| **Flan-UL2** (20B) | 50.3 | 0.321 | 0.378 |
| **OPT-IML-M-S** (1.3B) | 51.0 | 0.254 | 0.293 |
| **OPT-IML-M-L** (30B) | 50.7 | 0.266 | 0.312 |
| **GPT-3-D3** (175B) | 50.0 | 0.435 | 0.494 |
| **GPT-3-D2** (175B) | 45.7 | 0.310 | 0.354 |
| **Stable Vicuna** (13B) | 42.7 | **0.189** | **0.240** |

Table 3: Inference and Human Alignment Performances of LLMs on HighChaosNLI. The model categorizations and estimation methods are the same as Table 2. All the outputs are averaged over three runs.

causes of this phenomenon by spanning out the entropy distribution. On the consistent finding that GPT-3-D3 performs worse than Flan-T5-XXL in solving NLI tasks (Table 2) and capturing human disagreement levels (Figure 3), even in the High-ChaosNLI dataset (Table 3), as can be observed in Figure 4, GPT tends to be more overconfident, showing a entropy of less than 0.1 in most samples. In contrast, the human entropy is mostly evenly distributed in the range of 0.4 to 0.6 for ChaosNLI-$\alpha$ and 0.8 to 1.0 for ChaosNLI-S/M. On the other hand, Flan-T5-XXL exhibits lower confidence than GPT-3-D3 but higher confidence than humans, and Stable Vicuna is uncertain in most instances.

**Effect of Varying Prompts** To observe the effect of prompt sensitivity on varying prompt templates, we craft variations of the pre-selected prompt. For SNLI and MNLI, we sample out five prompt vari-

ants from the Flan repository[14] and make sensible variants for ChaosNLI-$\alpha$ as it is not part of the Flan mixture. From Table 4, it is shown that the prompt variation generally benefits Flan models in the ChaosNLI-S/M datasets as they were exposed to the prompt templates. However, the pre-selected single prompt is beneficial in performance for the ChaosNLI-$\alpha$ dataset for Flan models and all datasets in GPT-3-D3 and Stable Vicuna. The performance drop using prompt variation is even more severe for GPT-3-D3, suggesting the preferred usage of a carefully crafted single prompt over using unseen input templates. However, this does not mean that the single prompt should always be preferred since variations of prompts may display fairer performance trends of diverse models within the ground of robustness.

**What causes LLMs to disagree?** Sources of human disagreement have been well studied, but there is a lack of study of the disagreement sources for LLMs. We try to find the causes of LLM disagreements by drawing a relationship between LLM entropy level and human disagreement sources (discussed in Jiang and Marneffe, 2022) for each sample (Figure 5). However, no visible correlation of LLM entropy on human entropy is displayed across identified sources of human disagreement. This suggests that the cause of LLM disagreements may be due to factors other than human entropy and disagreement sources. Thus, Under the naive assumption that LLMs will attend to similar cues to humans, we are not fully uncovering the lens of why LLMs truly disagree.

---

[14]https://github.com/google-research/FLAN/

| Model | ChaosNLI-$\alpha$ | | | ChaosNLI-S | | | ChaosNLI-M | | |
|---|---|---|---|---|---|---|---|---|---|
| | Acc↑ | JSD↓ | DCE↓ | Acc↑ | JSD↓ | DCE↓ | Acc↑ | JSD↓ | DCE↓ |
| **Flan-T5-L** (780M) | **72.9** | **0.228** | **0.255** | 54.6 | 0.341 | 0.382 | 59.7 | **0.299** | **0.325** |
| w/ Prompt Variation | 64.9 | 0.279 | 0.317 | **63.3** | **0.303** | **0.330** | **64.7** | 0.303 | 0.331 |
| **Flan-T5-XL** (3B) | **83.2** | **0.166** | **0.171** | 71.8 | 0.255 | 0.264 | 64.5 | 0.272 | 0.304 |
| w/ Prompt Variation | 81.6 | 0.184 | 0.193 | **73.8** | **0.231** | **0.243** | **68.3** | **0.271** | **0.297** |
| **Flan-T5-XXL** (11B) | **84.9** | **0.159** | **0.154** | 67.9 | 0.270 | 0.289 | **72.6** | 0.271 | 0.293 |
| w/ Prompt Variation | 83.8 | 0.162 | 0.164 | **68.7** | **0.259** | **0.279** | 71.6 | **0.260** | **0.285** |
| **GPT-3-D3** (175B) | **76.5** | **0.254** | **0.249** | **62.7** | **0.348** | **0.374** | **63.3** | **0.373** | **0.402** |
| w/ Prompt Variation | 72.3 | 0.285 | 0.29 | 50.1 | 0.402 | 0.453 | 51.5 | 0.403 | 0.452 |
| **Stable Vicuna** (13B) | **55.6** | **0.310** | **0.368** | **34.2** | **0.390** | **0.451** | **45.4** | **0.303** | **0.342** |
| w/ Prompt Variation | 51.4 | 0.324 | 0.386 | 29.9 | 0.431 | 0.503 | 42.4 | 0.337 | 0.382 |

Table 4: Inference and Human Alignment Performances on the ChaosNLI Datasets with and without Prompt Variations. The estimation methods for each model are the same as Figure 4. All the outputs are averaged over three runs, and bold texts indicate the best value for each model and column.

## 6 Discussion

**LLMs do not perform well in NLI.** Despite minimal, unknown, or absence of exposure to the NLI task, we anticipated that state-of-the-art LLMs such as GPT-3 and Stable Vicuna could reason with this relatively basic inference problem. The models are trained with billions of parameters and are known to be effective in helping real-world users solve diverse, complex tasks (Ouyang et al., 2022). However, the unforeseen poor performance of these models casts doubt as to whether they possess true general language understanding abilities.

The problem is exacerbated for distilled models (*e.g.* Stable Vicuna) that are fine-tuned using proprietary LLMs, a performance discrepancy issue similarly raised by Gudibande et al., 2023. Since smaller LLMs fully or partially trained with NLI tasks could perform much better than the MUE and NE models, this hints at a task-specific latent factor in NLI tasks where supervised training is beneficial and required for a wider definition of natural language understanding. In fact, as these LLMs can simply be fine-tuned to perform better for NLI tasks, a stricter evaluation criterion is needed to assess the genuine understanding capability of LLMs.

**Characterizing Disagreement with respect to Ambiguity and Uncertainty** Previous studies relate multiple annotations not only to disagreement (Uma et al., 2021; Gordon et al., 2021), but also to ambiguity (Min et al., 2020; Tamkin et al., 2022; Liu and Liu, 2023), and mostly to uncertainty (Fox and Ülkümen, 2011; Xiao and Wang, 2021; Kuhn et al., 2022; Zhan et al., 2023; Hu et al., 2023). The

definitions of ambiguity, uncertainty, and disagreement have the potential to be conflated and disambiguated. In our paper, we use the multinomial soft label estimate of a model as a representation of "disagreement". When estimating this distribution with MCE, our modeling assumption treats each query to the model is analogous to asking an individual annotator to provide a label. In contrast, LPE is analogous to asking an individual to assign the scores to each option. Whereas most works exploit disagreement or uncertainty to improve various NLP task performances (Zhang et al., 2021; Fornaciari et al., 2021b; Yu et al., 2022; Zhou et al., 2023) our study focuses on evaluating the models. We find that using both methods for estimating the multinomial label distribution by querying the language model are not calibrated well with the human annotations.

**Other domain tasks are transferable to NLI.** Our work can be expanded to test LLMs on other NLP applications (Plank, 2022) such as Question Answering (De Marneffe et al., 2019), Fact Verification (Thorne et al., 2018), and Toxic Language Detection (Schmidt and Wiegand, 2017; Sandri et al., 2023). Further, our method can be applied for tasks that contain disagreements since they are easily transferable to NLI tasks (Dagan et al., 2006) like the QNLI dataset from Table 2, for example, instead of directly asking controversial questions (*e.g.*, abortion) to the model (Santurkar et al., 2023), the question format can be modified into a declarative statement in the premise and place a possible answer in the hypothesis with a binary True/False

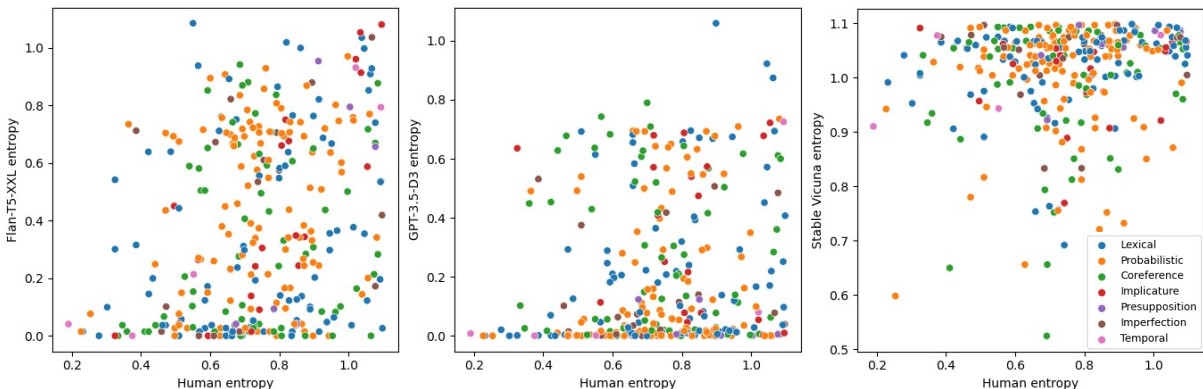

Figure 5: Relationship between Human and LLM Entropy Levels Divided with Different Human Disagreement Reasons. The estimation methods for each model are the same as Figure 4.

label (Dagan et al., 2006). Thus, if these complicated tasks can be formulated in a way where the LLM can estimate a multinomial distribution over a set of classes, our methods are applicable.

However, we should consider the target tasks when tracing "human disagreement" only when it is a significant signal that needs to be captured. For example, since it is important to include diverse opinions, we can easily apply our methods to detect disagreements in hate speech (Schmidt and Wiegand, 2017). In contrast, spotting disagreement in the arithmetic reasoning task (Cobbe et al., 2021) might be less important since it often requires a logical step-by-step reasoning procedure to obtain an accurate answer.

**How can we better align LLMs to represent dissenting voices?** We point out the current limitation of utilizing LLMs to represent a larger human population, especially when disagreements are present. The causes of this phenomenon are indiscernible due to the entanglement of miscalibration of out-of-distribution (OOD) inference, additional noise due to disagreement and ambiguity, prompt sensitivity, and more aspects that are yet to be identified. Even though simple remedies of temperature scaling (Ackley et al., 1985; Wang et al., 2022), incorporating logit bias, constrained decoding (Ziems et al., 2023), or direct supervision to multiple annotations (Zhang et al., 2021; Fornaciari et al., 2021a) might mitigate the misalignment, these methods are unrealistic and not scalable due to the exhaustive hyperparameter tuning and additional data collection required to represent the population of interest.

However, as LLM applications are becoming more ubiquitous, it is important for them to faithfully represent a larger population, preferably including the voices of minorities. Thus, we suggest that future LLMs could be improved to reflect human disagreements in diverse means, for example, by fine-tuning with ambiguous instances (Liu et al., 2023). As LLMs are shown to be aware of their ignorance (Kadavath et al., 2022) and have the ability to express their level of confidence (Lin et al., 2022; Zhou et al., 2023), we expect future works to address similar approaches in the aspect of alignment towards the human disagreement distribution. In this way, the reconstructed model distribution with MCE and LPE may better capture different interpretations from human individuals, aiding accountability.

## 7 Conclusions

In this paper, we compare the performance of instruction-following generative LLMs with other fully fine-tuned smaller models on the fundamental NLI task. First, by experimenting on four different NLI datasets, we show LLMs are not performing well in the NLI task, considering their touted language comprehension capabilities. Further, in agreement with the need for multiple annotations for disagreeable NLP tasks, LLMs also fail to align with human disagreements in the ChaosNLI and PK2019 datasets. Additional development is needed to capture representative human distributions, as well as to discover key factors to disagreement sources that can influence the LLM's answer distribution.

## Limitations

This work shows the limited ability of billion-scale LLMs in inference and disagreement tasks. Al-

though we test with the dataset annotated with numerous human subjects per sample, 100 people may not be enough to represent the human disagreement distribution well. After more releases of human label variation datasets, our study can be extended by covering a wider range of model types and creating evaluation benchmarks to measure the degree of disagreement. If we have robust LLMs in inference and disagreement, we could then try to find the latent factors that might not be human-interpretable but lead to disagreement in LLMs and compare them with those of humans.

## Ethics Statement

As our work directly employs trained large language models without any extra process of fine-tuning, the risks and potential biases incurred by the model checkpoints (*e.g.*, dataset selection, training configurations) remain the same as the original works.

## Acknowledgments

This work was supported by Institute of Information & communications Technology Planning & Evaluation (IITP) grant funded by the Korea government (MSIT) (No.2019-0-00075, Artificial Intelligence Graduate School Program (KAIST)) and Artificial intelligence industrial convergence cluster development project funded by the Ministry of Science and ICT (MSIT, Korea) & Gwangju Metropolitan City.

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

| Model | Precision | Acc | JSD |
|---|---|---|---|
| Flan-T5-XXL | FP32 | 75.4 | 0.233 |
| | BF16 | 75.1 | 0.233 |

Table 5: Effect of Precision on Inference and Human Alignment Performances for Flan-T5-XXL. The distribution is estimated using MCE (OS), same as Table 2. The outputs are averaged over three ChaosNLI sub-datasets.

## A Hyperparameters

Generally, we try to set similar hyperparameters to all the models with some exceptions due to model performance and/or cost issues.

**Temperature** To scale the confidence of the generated output in a post-hoc manner, we unify the temperature to be 1 (*i.e.*, no scaling). There exist other precedents that use a smaller temperature for a more deterministic output (Santurkar et al., 2023) or compare outputs of models with varying temperatures (Ouyang et al., 2022). However, as we jointly assess LLMs on the accuracy of NLI and human disagreement alignment, we argue that having a fixed, un-scaled temperature to generate model outputs better aligns with our research goal of estimating model outputs to capture human disagreement distribution.

**Generation Length** Easily adjustable by all APIs, including OpenAPI and Huggingface, we have varying generation lengths per prompt design. As discussed in Section 4.3, NS is a cost-efficient alternative method for OS, solely needing a single output token of numbers. Thus in LPE, a method for single token probability output, we only use the OS prompt for effective token probability calculation. We set a maximum token output length of 10 for MCE and 1 for LPE.

**Floating Point** We load models of size greater than 10 billion parameters (except for GPT-3) with half the precision (bfloat16; BF16). We observe Flan-T5-XXL shows a negligible increase in performance when using the original precision (single-precision floating-point; FP32) (Table 5).

## B Levels of NLI Exposure

We outline the level of exposure to the NLI task for each model since it is an influential factor that affects the accuracy and human-alignment performances of the models.

### B.1 Full Exposure (FE) Models

The models below are fine-tuned with the training set of an NLI task as outlined in Nie et al., 2020b.

- Models: BERT and RoBERTa

### B.2 Partial Exposure (PE) Models

These models are partially exposed to the NLI task in the fine-tuning stage. However, the extent of exposure is different by the adopted fine-tuning strategy, thus listed in decreasing order.

#### Flan Collection

- Models: Flan-T5 models and Flan-UL2
- The Flan Collection (Longpre et al., 2023) is a collection of datasets in the format of instructions to enable generalization to diverse unseen tasks. It employs a fine-tuning strategy of a maximum of 1836 NLP tasks with some NLI tasks taken into account (*e.g.*, ANLI, RTE, MNLI, QNLI, SNLI, etc.).[15]

#### Instruction Meta-Learning (IML) Bench

- Models: OPT-IML-M models
- Instruction Meta Learning (IML) Bench (Iyer et al., 2022) is a more common benchmark that uses 1500+ NLP tasks in the fine-tuning stage. Flan is a major portion of this benchmark, with other major portions in other large datasets. We expect some NLI exposure but not as strong as the models fine-tuned by the Flan dataset.

### B.3 Minimal/Unknown Exposure (MUE) Models

The models below are unknown to the extent of exposure to a specific NLI task.

- Models: GPT-3-D2, GPT-3-D3

The InstructGPT paper (Ouyang et al., 2022) does elaborate that the models utilizes a reward model in the process of RLHF (Reinforcement Learning from Human Feedback), and it is fine-tuned by a variety of NLP datasets, including MNLI. However, the serviced models are not directly mapped to the models of the paper, leaving the exposure to NLI largely unknown[16].

[15]https://github.com/google-research/FLAN/tree/main
[16]Refer to the OpenAI Documentation

### B.4 No Exposure (NE) Models

The below model does not have any exposure to a specific NLI task.

- Model: Stable Vicuna

## C Postprocessing

Unlike conventional approaches of fine-tuning models directly on the downstream NLI dataset, one of the challenges in assessing an NLI task is the variability of generated outputs. To transform and choose valid options from the generated outputs, we conduct postprocessing through a manually crafted dictionary for each option (*See* Valid option examples on the last page.).

## D Distribution Alignment Among LLMs

We illustrate the averaged sample-level JSD entropy for each model pair (Figure 6) to visualize the trend of alignment among LLMs. Throughout all four JSD distribution plots, the scale and range of the JSD values differ for each data. Still, the general trend is maintained, where ChaosNLI-$\alpha$ shows low JSD values overall, likely attributed to lower task difficulty witnessed by the performance gap among datasets in Table 2. The best-performing models, Flan-T5-XXL and Flan-UL2, present the lowest disagreement in entropy for all plots.

Although the size and type of model are influencing factors, the most consistent factor is the type of instruction fine-tuning introduced for each model. Throughout all plots, the alignment is well shown for the group of models fine-tuned by the Flan dataset and the IML Bench. As we expect more research in the scope of human alignment in NLP, the evaluation of the human alignment among the models with the same fine-tuning process can also be studied and reported.

However, a strong distinction needs to be made in which an overall lower number of JSD values in this plot does not mean that a model has always had a good performance in human disagreement alignment. This figure merely delineates the alignment trends among models.

## E Effect of Few-Shot Examples

We observe no consistent benefit nor harm in experimenting with few-shot settings that resemble the human annotation process more than zero-shot settings (Table 6). In fact, zero-shot evaluation generally seems to show better performances across

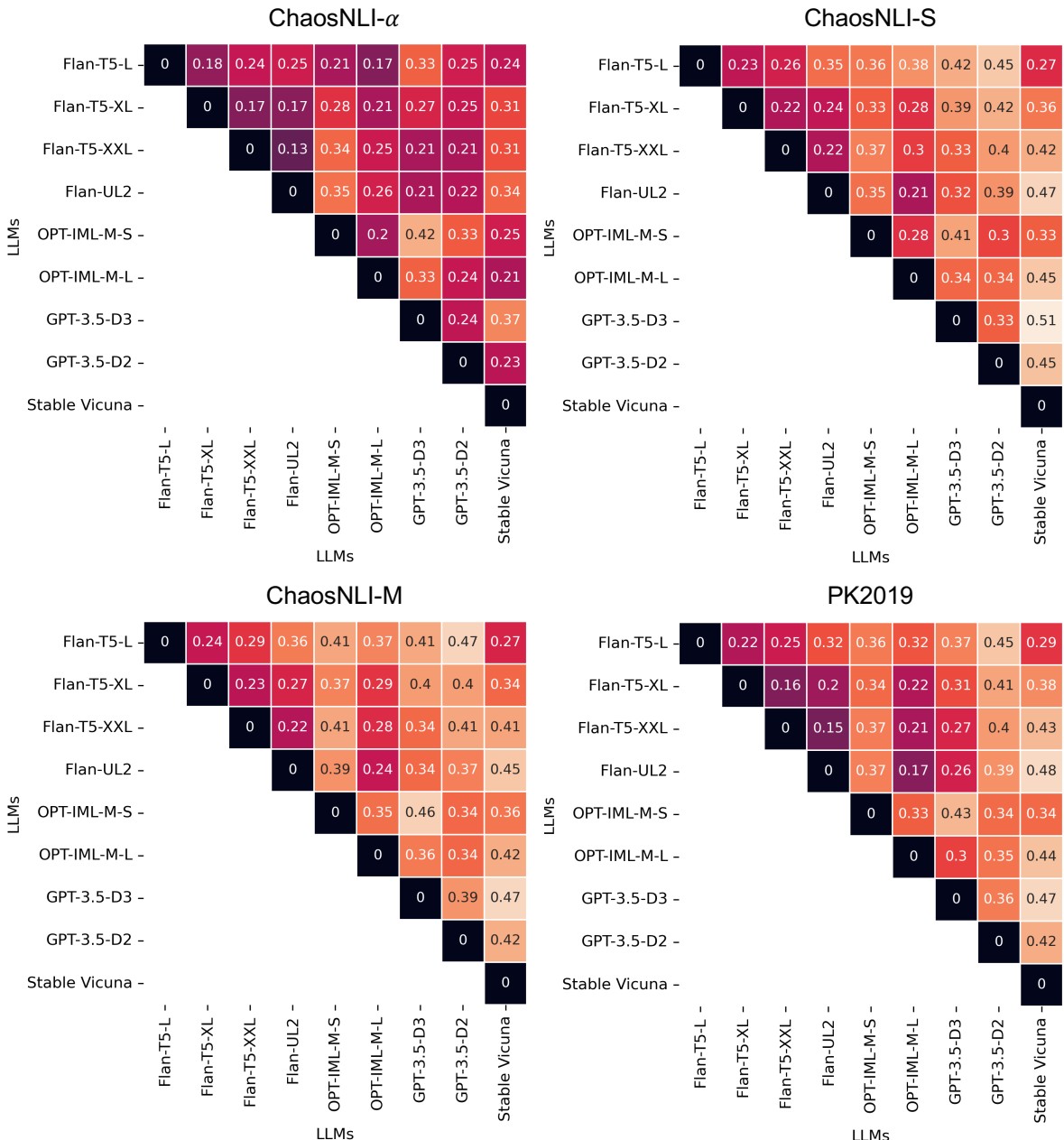

Figure 6: JSD Distribution between All Combinations of Pairs for LLMs. The darker plot indicates a similar distribution between a pair of models. The estimation methods for each model are the same as Table 2.

datasets and models compared to the few-shot evaluations. In the case of Stable Vicuna, the performance increases in the 1-shot setting for $\alpha$-NLI and the 3-shot setting for SNLI. However, we notice a plunge in 5-shot performance, especially for the MNLI dataset.

## F  Prompt Examples

We present examples of prompts we used during the generation process in Figure 1 (*See* two prompt examples on the last page.). We incorporate a sug-

gested general prompt template pre-specified for a specific model. For example, we implement a human and assistant-style prompt template for Stable Vicuna. Otherwise, we leave the template format the same for the rest of the models.

| Model | ChaosNLI-$\alpha$ | | | ChaosNLI-S | | | ChaosNLI-M | | |
|---|---|---|---|---|---|---|---|---|---|
| | Acc↑ | JSD↓ | DCE↓ | Acc↑ | JSD↓ | DCE↓ | Acc↑ | JSD↓ | DCE↓ |
| **Flan-T5-XXL (0 Shot)** | **85.0** | 0.160 | 0.155 | 67.3 | **0.271** | **0.291** | 72.6 | 0.269 | 0.290 |
| **+ 1 Shot** | 84.9 | 0.159 | **0.154** | **68.0** | 0.278 | 0.296 | **74.8** | **0.261** | **0.278** |
| **+ 3 Shot** | 83.6 | 0.163 | 0.160 | 67.0 | 0.285 | 0.304 | 74.0 | 0.271 | 0.290 |
| **+ 5 Shot** | 84.9 | **0.158** | **0.154** | 65.7 | 0.288 | 0.309 | 73.2 | 0.275 | 0.295 |
| **GPT-3-D3 (0 Shot)** | 76.1 | 0.254 | 0.249 | **62.4** | **0.348** | **0.374** | **63.0** | **0.376** | **0.405** |
| **+ 1 Shot** | 77.7 | 0.240 | 0.235 | 6.2 | 0.400 | 0.433 | 61.1 | 0.414 | 0.445 |
| **+ 3 Shot** | 80.7 | **0.233** | 0.216 | 55.7 | 0.407 | 0.442 | 58.2 | 0.436 | 0.471 |
| **+ 5 Shot** | **81.7** | **0.233** | **0.213** | 57.9 | 0.396 | 0.426 | 62.2 | 0.425 | 0.454 |
| **Stable Vicuna (0 Shot)** | 55.7 | 0.310 | 0.368 | 33.5 | 0.391 | 0.454 | **46.2** | **0.304** | **0.342** |
| **+ 1 Shot** | **64.3** | **0.290** | **0.336** | 38.6 | 0.377 | 0.436 | 41.7 | 0.311 | 0.355 |
| **+ 3 Shot** | 56.2 | 0.296 | 0.346 | **43.4** | **0.351** | **0.405** | 32.8 | 0.352 | 0.418 |
| **+ 5 Shot** | 56.1 | 0.298 | 0.350 | 35.8 | 0.379 | 0.441 | 28.8 | 0.370 | 0.441 |

Table 6: Inference and Human Alignment Performances on the ChaosNLI Datasets for Zero-shot and Few-shot Settings. The estimation methods for each model are the same as Table 2. Bold texts indicate the best value for each model and column.

## Valid option examples for ChaosNLI-$\alpha$/S/M and two prompt types - OS and NS

```
dict_alphanli_OS = {'1' : ['1','Hypothesis 1',...]
                    '2' : ['2','Hypothesis 2',...]}

dict_alphanli_NS = {'1' : '1'
                    '2' : '2'}

dict_s&mnli_OS = {'e' : ['entail','infer','yes', ...]
                  'c' : ['contradict','oppose','no', ...]
                  'n' : ['neutral','unanswerable',...]}

dict_s&mnli_NS = {'e' : '1'
                  'c' : '2'
                  'n' : '3'}
```

## Prompt example for ChaosNLI-$\alpha$ using OS

**INPUT**

Read the following and determine if the hypothesis can be inferred from the premise.
Observation Start: My roommates put up their Christmas tree this year.
Observation End: This is what it's like living with a cat.
Hypothesis 1: The roommates soon had to take the tree down.
Hypothesis 2: The cat enjoyed the ornaments and garland and slept under the tree.
Options: Hypothesis 1, Hypothesis 2

- - - - - - - - - - - - - - - - - - - - - - - - - - - - - - - - - - - - - - - - -

**OUTPUT**

Answer: <Generated Output>

## Prompt example for ChaosNLI-S/M using NS

**INPUT**

Read the following and determine if the hypothesis can be inferred from the premise.
Premise: This town, which flourished between 6500 and 5500 b.c. ... appear on Anatolian kilims.
Hypothesis: This town is over 8000 years old.
Options: 1: entailment, 2: contradiction, 3: neutral

- - - - - - - - - - - - - - - - - - - - - - - - - - - - - - - - - - - - - - - - -

**OUTPUT**

Answer: <Generated Output>