# OpenReview forum: "Can Large Language Models Capture Dissenting Human Voices?"
_EMNLP/2023/Conference — EMNLP 2023 Main_

### Official Review · Reviewer_UZ9d · 2023-07-27

**Typos Grammar Style And Presentation Improvements:** 116
**Soundness:** 3

**Excitement:**

4: Strong: This paper deepens the understanding of some phenomenon or lowers the barriers to an existing research direction.

**Missing References:**

Uma, Alexandra N., et al. "Learning from disagreement: A survey." Journal of Artificial Intelligence Research 72 (2021): 1385-1470. https://jair.org/index.php/jair/article/view/12752

Tommaso Fornaciari, Alexandra Uma, Silviu Paun, Barbara Plank, Dirk Hovy, and Massimo Poesio. 2021. Beyond Black & White: Leveraging Annotator Disagreement via Soft-Label Multi-Task Learning. In Proceedings of the 2021 Conference of the North American Chapter of the Association for Computational Linguistics: Human Language Technologies, pages 2591–2597, Online. Association for Computational Linguistics. https://aclanthology.org/2021.naacl-main.204/

Shujian Zhang, Chengyue Gong, and Eunsol Choi. 2021. Learning with Different Amounts of Annotation: From Zero to Many Labels. In Proceedings of the 2021 Conference on Empirical Methods in Natural Language Processing, pages 7620–7632, Online and Punta Cana, Dominican Republic. Association for Computational Linguistics. https://aclanthology.org/2021.emnlp-main.601/

Yuxia Wang, Minghan Wang, Yimeng Chen, Shimin Tao, Jiaxin Guo, Chang Su, Min Zhang, and Hao Yang. 2022. Capture Human Disagreement Distributions by Calibrated Networks for Natural Language Inference. In Findings of the Association for Computational Linguistics: ACL 2022, pages 1524–1535, Dublin, Ireland. Association for Computational Linguistics. https://aclanthology.org/2022.findings-acl.120/

Yuxin Xiao, Paul Pu Liang, Umang Bhatt, Willie Neiswanger, Ruslan Salakhutdinov, and Louis-Philippe Morency. 2022. Uncertainty Quantification with Pre-trained Language Models: A Large-Scale Empirical Analysis. In Findings of the Association for Computational Linguistics: EMNLP 2022, pages 7273–7284, Abu Dhabi, United Arab Emirates. Association for Computational Linguistics. https://aclanthology.org/2022.findings-emnlp.538/

**Paper Topic And Main Contributions:**

Evaluation of several recent Large Language Models (LLMs) on several Natural Language Inference (NLI) benchmarks, in particular benchmarks designed to capture human disagreements.

- Models are prompted on multiple-choice NLI in two settings, Option Selection (OS) and Number Selection (NS).
- Two methods are proposed to estimate the model output distribution, Monte Carlo Estimation (MCE) and Log Probability Estimation (LPE), which can be used in different situations depending on whether the model exposes the estimated probabilities of its output tokens.
- Evaluation metrics include accuracy as well as distribution differences between estimated LLM output and human labels, using three metrics, Jensen-Shannon Divergence (JSD), Kullback-Leibler Divergence (KL) and Distribution Calibration Error (DCE).

Results show that larger models with no exposure to NLI data underperform and do not align with human label distributions.

**Questions For The Authors:**

Question A: in line 184 you mention the unknown non-constant probability mass in LPE. Is this a limitation? How does it affect the conclusions one can draw from this metric? Is there a way to mitigate it?

Question B: are the encoder-only models (BERT, RoBERTa) also prompted in the same ways as the encoder-decoder (Flan-T5) and decoder-only (GPT, OPT, Vicuna) models? The usual methodology is to fine-tune them with a classification layer, but from the paper it seems they are prompted. Is there previous work comparing these approaches?

Question C: which distribution estimation is used for the experiments reported in Figure 3?

**Reasons To Accept:**

Capturing human disagreements is an important open problem in LLMs, and this paper utilizes the extensive work done previously on human disagreements in NLI to evaluate LLMs on this problem.

The proposed output distribution estimation methods can be useful for future work estimating and calibrating LLMs.

The results highlight the limitations of current LLMs and identify factors that limit their accuracy and alignment with human label distributions in NLI.

**Reasons To Reject:**

[Addressed in rebuttal] Insufficient positioning with respect to previous work on label distribution and capturing human disagreements (see Missing References). While Distributed NLI [1] is cited in passing, for example, a more thorough comparison is warranted between it and the current approach, as it also proposes similar output label distribution estimation methods and evaluates them on NLI.

[Addressed in rebuttal] As acknowledged in line 480, the authors "formally define and represent the multinomial soft label estimate of a model as "disagreement," but there is no theoretical support to link these terminologies". I could not say it better, and I believe this is a serious limitations that was a major source of confusion for me while reading the paper: while clearly a form of anthropomorphism, it is not clear what "disagreement" refers to when used to describe LLMs - is it about disagreement between different LLMs or within the same LLM? From the paper it is evident that the latter is meant, but this metaphorical narrative obscures the real research question in a way that makes it difficult to assess its validity.

The framing of the paper suggests that the NLI task as operationalized in datasets bears some kind of "truth" that LLMs (with no exposure to NLI data) fail to capture despite their language capabilities. In reality, as acknowledged by both Manning [2] and Pavlick and Kwiatkowski, humans have different assumptions and follow different processes in NLI annotation, even if following a "natural" rather than a "prescriptivist" ask formulation. Especially where the "prescriptivist" approach is used for annotation, though, it is not fair to evaluate the "no-exposure" models and expect them to approximate humans who have read the (potentially artificial) task guidelines.

[Addressed in rebuttal] The characterization of models as having "full exposure", "partial exposure" or "no exposure" to NLI seems too coarse grained to me. While it may simply not be known to which extent GPT-3 was exposed to NLI data, there is a large difference between fine-tuning on one or thousands of examples. A more rigorous approach could be to apply few-shot learning and then compare the effect of exposure size with a single baseline model. In particular, the conclusion that "LLMs do not perform NLI like humans" is simply misleading, since human annotators almost certainly do see examples before or during the annotation process.

[1] Xiang Zhou, Yixin Nie, and Mohit Bansal. 2022. Distributed NLI: Learning to Predict Human Opinion Distributions for Language Reasoning. In Findings of the Association for Computational Linguistics: ACL 2022, pages 972–987, Dublin, Ireland. Association for Computational Linguistics. https://aclanthology.org/2022.findings-acl.79/

[2] Christopher D. Manning. 2006. Local textual inference: It’s hard to circumscribe.

**Reproducibility:**

2: Would be hard pressed to reproduce the results. The contribution depends on data that are simply not available outside the author's institution or consortium; not enough details are provided.

**Reviewer Confidence:**

3: Pretty sure, but there's a chance I missed something. Although I have a good feel for this area in general, I did not carefully check the paper's details, e.g., the math, experimental design, or novelty.

---

> ### Author Rebuttal · Authors · 2023-08-29
>
> Dear Reviewer UZ9d,
>
> Thank you for the time to review our paper thoroughly. We deeply appreciate how you found our methods useful for estimating LLMs, which currently lack accuracy and alignment with human disagreement in NLI.
>
> Below, we have tried our best to address your questions and concerns. Please take a look at our response and let us know if further clarification is needed.
>
> **Q1)  You mention the unknown non-constant probability mass in LPE. Is this a limitation? How does it affect the conclusions one can draw from this metric? Is there a way to mitigate it?**
>
> The unknown probability mass is displayed due to two reasons: a missing option or the model's characteristic to be unconfident. It is important this is carefully dealt with as the amount of disagreement can be represented as a part of the unknown mass as well.
>
> For the prior component, Santurkar et al. assign the probability mass of missing options to the minimum value of the remaining mass or the minimum probability assigned to any of K token choices and normalizes after, which will lead to an unsound overestimation of unspecified (i.e., missing) options. For Ziems et al., logit bias adjustments were also shown to be effective. In contrast, we make adjustments by not assigning probability mass for the missing options and directly normalizing our output which might be overestimating the probabilities of existing options, but refraining from any problematic assignment to missing options.
>
> For the latter component, the underconfidence of the LLM outputs is a major problem. While, numerous methods to re-calibrate the model confidence, through temperature scaling or label smoothing (Wang et al., 2022); however, this is not an option we adopt as we plan to investigate the unscaled evaluation across the model without optimizing it for every single task/model.
>
> **Q2) Are the encoder-only models (BERT, RoBERTa) also prompted in the same ways as the encoder-decoder (Flan-T5) and decoder-only (GPT, OPT, Vicuna) models? The usual methodology is to fine-tune them with a classification layer, but from the paper it seems they are prompted. Is there previous work comparing these approaches?**
>
> No, the encoder-only models are not prompted but evaluated by the usual method that you mentioned. Plus, this is why we separate Full Exposure (FE) models from Partial/No Exposure (PE/NE) models, but we should have been clearer. Additionally, as far as we are concerned, no previous approaches jointly compare encoder-only (FE models in this work), encoder-decoder, and decoder-based models in the scope of NLI and human disagreement. Thus, such comparisons can be considered as the main novelty introduced first in our paper in the era of generative LLMs which are too large for single-task fine-tuning.
>
> **Q3) Which distribution estimation method is used for the experiments reported in Figure 3?**
>
> We are aware of this confusion, mainly because of the mismatch of label shorthand for model names across figures and tables. We will first revise the label shorthand (e.g., Flan-T5-XXL (Table 1) and FXXL (Figure 3) $\rightarrow$ FlanXXL, GPT-3.5-D3 (Table 1) and G3 (Figure 3) $\rightarrow$ DAV3). In Figure 3, we use LPE with a prompt type of Number Selection (NS) for G3, G2, and SV, and  MCE with a prompt type of Option Selection (OS) for the other models, same as Table 3. We will add these details in the caption for Figure 3 to avoid confusion.
>
> **Insufficient positioning with respect to previous work on label distribution and capturing human disagreements (see Missing References)**
>
> Yes, we appreciate your comment on how our paper has insufficient positioning w.r.t. previous work. This is possibly due to our emphasis on mainly showing comprehensive, easily applicable evaluations for diverse tasks and models including generative LLMs on the alignment of human disagreement.  We will take on board your suggestion to discuss papers concerning uncertainty recalibration (Wang et al. 2022, Xiao et al. 2022), and disagreement (Uma et al., 2021, Fornaciari et al., 2021, Zhang et al. 2021). We will further position ourselves on these previous studies regarding label variation and human disagreements for the final version of the manuscript.
>
>
> **While Distributed NLI [1] is cited in passing, for example, a more thorough comparison is warranted between it and the current approach, as it also proposes similar output label distribution estimation methods and evaluates them on NLI.**
>
> Yes, Distributed NLI is a highly related work that we were paying attention to. However, as we previously mentioned our goal is to inspect general methods that are applicable to different models of size (Encoder, Decoder, Encoder-Decoder) and openness (Huggingface, Closed API such as OpenAPI, etc.), We try to evaluate the simple methods that we suggest without calibrating the confidence on the model for every model to respective datasets.
>
>
> **While clearly a form of anthropomorphism, it is not clear what "disagreement" refers to when used to describe LLMs - is it about disagreement between different LLMs or within the same LLM? From the paper, it is evident that the latter is meant, but this metaphorical narrative obscures the real research question in a way that makes it difficult to assess its validity.**
> Thank you for confirming that our methods only regard the agreement coming from within the same LM. Our intention for the initial titling comes from how LLMs are capable of representing dissenting voices through the lens of NLI. With such scaling parameters and datasets, which include voices of a significant population, we originally hypothesized LLMs to be able to represent disagreement, which we found not to be true in our experiments.
>
>
> **The characterization of models as having "full exposure", "partial exposure" or "no exposure" to NLI seems too coarse grained to me.**
>
> We strongly agree with your comment on the coarse-grained categorization. In our initial stage of the research, the Partial Exposure (PE) level was divided into PE and Minimal Exposure (ME), where ME is assigned to GPT-3.5-D3 because its reward model was fine-tuned with the MNLI dataset. This way, GPT-3.5-D3 was differentiated from FLAN and OPT-IML models, which might explain our first result (lines 282-283) better. We will take your valuable comment to consider more fine-grained categorization of models.
>
> **A more rigorous approach could be to apply few-shot learning and then compare the effect of exposure size with a single baseline model. In particular, the conclusion that "LLMs do not perform NLI like humans" is simply misleading, since human annotators almost certainly do see examples before or during the annotation process.**
>
> We appreciate your concern about the unfairness of the evaluation for LLMs without any given instances. However, we do not try to align LLMs with the same procedure with human annotators as we are focused more on the representation of human dissent, though maybe confounded with our title. Nevertheless, we are very interested in your concern, and it will be interesting to provide the “prescriptivist” approach evaluation with additional experimental results in the future study.
>
> We also deeply appreciate you for the comments on missing references and typos/presentation improvements to bring a stronger presentation for the paper. We will respond to some of them:
>
> **Missing References:**
>
> We will integrate the following important references in our Related Works section to clear our position:
> •	Comprehensive survey on the need to represent disagreement in various tasks in NLP & CV (Uma et al., 2021)
>
>
>
> •	Incorporates *disagreement* as an informative signal in training models and uses the multinomial distribution to train the models  (Fornaciari et al., 2021)
>
>
>
> •	Introduces a novel annotating scheme to assign multiple labels for each instance, which can be used as high-quality datasets to better train models to achieve higher accuracy and lower JSD scores. (Zhang et al. 2021)
>
>
>
> •	Uses calibration methods such as label smoothing and temperaturing scaling to lower divergence scores between model probability and true human opinion distribution. (Wang et al. 2022)
>
>
>
> •	Introduces major features that better the pipeline of calibration error minimization expected calibration error (ECE) as a main metric to quantify uncertainty. (Xiao et al. 2022)
>
> **Typos Grammar Style And Presentation Improvements:**
>
>
>
>
> •	**[Additional Explanation]** Thank you for pointing this out.  DCE is a total variation distance measure between LLM distribution estimated with MCE/LCE and human disagreement distribution (Baan et al., 2022). This metric is different from widely used Expected Calibration Error (ECE), which can often result in high ECE even if the model can perfectly estimate the human judgement distribution (Baan et al., 2022). We will add this explanation in the Related Works section (line 131).
>
>
>
> •	**[Sentence Revision]** We will revise the sentence starting in line 141 to the following sentence: For example, GPT-3.5 is fine-tuned to follow instructions with human feedback using reinforcement learning method (Ouyang et al., 2022).
>
>
>
> •	**[Clarification on Model Names]** GPT-3.5-D2/D3 refers to Text-Davinci-002/3, which is different from GPT-3.5-turbo. We exclude GPT-3.5-turbo in our experiment since at the time of experimentation it does not provide token-level log probability unlike Text-Davinci-002/3. We will make this clearer by changing the shorthand of GPT-3.5-D2/D3 to GPT-3-D2/D3.
>
>
>
>
> •	**[Miscitation]** Thank you for your correction for the miscitation on StoryCloze. We will change the original citation to the citation that you suggested (Mostafazadeh et al., 2016).
>
>
>
>
> •	**[Table 1 Clarification]** In Table 1 caption, clarify what "old to new labels" means. $\rightarrow$ We will clarify that there exists two labels (old and new) for each instance in ChaosNLI datasets. The old labels are the majority class options selected by only five annotators, a strategy selected for MNLI, SNLI, and $\alpha$-NLI datasets. The new labels are the partially revised ground-truth labels, by introducing 100 annotators for the ChaosNLI datasets.
>
>
>
> •	**[Figure 2 Revision]** Figure 2 should be more explicit about how the full prompt looks and how the output from the model is parsed and mapped to labels. $\rightarrow$ Yes, we should have been more explicit on the actual full prompts we used in Figure 2. Combining prompt examples in Appendix (p. 14) for different types of datasets and prompts, and parts of Figure 1 showing how the model outputs are sometimes discarded if they are not valid options or kept for vice versa might be needed to gain a better understanding on the process of generating LLM outputs.
>
> **References:**
>
>
> •	Santurkar, S. et al. (2023). Whose Opinions Do Language Models Reflect? In International Conference on Machine Learning. pages 29971-30004.
>
>
> •	Ziems et al., (2023). Can Large Language Models Transform Computational Social Science?. arXiv e-prints, arXiv-2305.03514.
>
>
> •	Zhou, X., Nie, Y., & Bansal, M. (2022). Distributed NLI: Learning to Predict Human Opinion Distributions for Language Reasoning. In Findings of the Association for Computational Linguistics: ACL 2022, pages 972-987.
>
>
> •	Wang, Y. et al. (2022). Capture human disagreement distributions by calibrated networks for natural language inference. In Findings of the Association for Computational Linguistics: ACL 2022, pages. 1524-1535.
>
>
> •	Nie, Y., Zhou, X., & Bansal, M. (2020). What Can We Learn from Collective Human Opinions on Natural Language Inference Data?. In Proceedings of the 2020 Conference on Empirical Methods in Natural Language Processing, pages 9131-9143.
>
>
> •	Baan, J., Aziz, W., Plank, B., & Fernández, R. (2022). Stop Measuring Calibration When Humans Disagree. In Proceedings of the 2022 Conference on Empirical Methods in Natural Language Processing, pages 1892-1915.
>
>
> •	Ouyang, L. et al. (2022). Training language models to follow instructions with human feedback. In Advances in Neural Information Processing Systems, Volume 35, pages 27730-27744.
>
>
> •	Mostafazadeh, N. et al. (2016). A corpus and cloze evaluation for deeper understanding of commonsense stories. In Proceedings of the 2016 Conference of the North American Chapter of the Association for Computational Linguistics: Human Language Technologies, pages 839-849.

---

### Official Review · Reviewer_DgQC · 2023-08-04

**Soundness:** 3

**Excitement:**

3: Ambivalent: It has merits (e.g., it reports state-of-the-art results, the idea is nice), but there are key weaknesses (e.g., it describes incremental work), and it can significantly benefit from another round of revision. However, I won't object to accepting it if my co-reviewers champion it.

**Missing References:**

Includes results on Internal Agreement from LLMS:
https://arxiv.org/abs/2303.15056

This work seems to solve the challenge of options not occurring in the top-k tokens using OpenAI's logit bias, see Table 1:
https://arxiv.org/pdf/2305.03514.pdf

An example of how minor wording shifts could be used to more accurately model different perspectives and how these effect model uncertainty
https://arxiv.org/pdf/2302.13439.pdf

A strongly motivated work looking at how model training leads to poor calibration with human perspectives
https://dl.acm.org/doi/abs/10.1145/3411764.3445423


**Paper Topic And Main Contributions:**

This work looks at LLM performance on the classic task of NLI and evaluates metrics which estimate the uncertainty of LLM models for these fixed label tasks. It then uses these metrics to look at whether model size, exposure, and architecture has significant effects on both model performance and the agreement between model uncertainty and human uncertainty. From this they drive towards surrounding whether or not LLMs are accurate models of human disagreement and uncertainty using NLI as a clear testbed that has a format which is broadly applicable.

**Questions For The Authors:**

A) On Lines 154-156, you cite the decoding strategy as a limiting factor for exploring the whole space. How does this apply to your NLI tasks, where the model is selecting between a finite number of options rather than performing open-ended decoding? With lines 162-163, it seems like you are saying each class label will be inferred based on the open-ended generation, but this isn't totally clear.

B) Line 182 - 185, suggest the probability mass is non-constant, but this is a choice by this calculation. Why don't you use the adjustment from Santurkar et al. 2023 in your citations to make the probability mass constant by estimating the mass of missing options as their upperbound?

E) The FLAN Models use the full ANLI and QNLI training datasets as part of their instruction fine tuning. Why are they marked as Partial Exposure?

D) Line 245-246, How can you confidently say that Davinci 2 has no exposure to these tasks? It received supervised finetuning on an unknown dataset, which based on my understanding means it could have partial or complete exposure to these tasks.

E) Line 264-268, Why can't you compare the probabilities of the first token of each option for OS LPE? Entailment, Contradiction, and Neutral all have different leading tokens so the probability of each leading token seems a valid proxy for the whole word.



**Reasons To Accept:**

- Understanding the mechanisms we can use to model human disagreement with LLMs is increasingly important as other works aim to use LLMs as annotators, models of human opinion, and even to score other ML systems. The focus of this work on developing a sound metric of mapping LLMs to distributions of opinions has potential impacts in each of those domains.

- The selection of models is extremely thorough, a significant positive aspect of this work over similar prior works of this kind.

**Reasons To Reject:**

- The fundamental comparison seems to incompletely answer the question layed out in the title, confounded by the way the models are compared to human annotators. The probability distribution of labels assigned by different human annotators is the average of opinions from multiple agents. However, disagreement does not neccesarily mean that each individual annotator was uncertain in the label they selected. Given the high variance from prompting, it seems not sufficiently sound to compare "agreement" of LLMs by using the sampled variance over one prompt.

- The paper does not clearly communicate actionable insights from their work. Santurkar et al. looks at model agreement to understand how models reflect society to provide actionable insights about the biases they may capture. Gillardi et al. looks at model agreement to understand whether uncertainty can be a proxy for annotation quality. Both of these works have already captured varying aspects of the overconfident single perspectives LLMs represent. This work has the potential to more systematically understand whether we can map model uncertainty to human uncertainty, but at present it misses that mark.

**Reproducibility:**

4: Could mostly reproduce the results, but there may be some variation because of sample variance or minor variations in their interpretation of the protocol or method.

**Reviewer Confidence:**

4: Quite sure. I tried to check the important points carefully. It's unlikely, though conceivable, that I missed something that should affect my ratings.

**Typos Grammar Style And Presentation Improvements:**

Lines 241-252: The naming conventions you use for davinci-002 and 003 are confusing if you are familiar with the api names. The first model officially referred to as GPT 3.5 in the official documentation is ChatGPT. Since here you aren't referring to ChatGPT, I'd remove the ".5" from your shorthand.

Line 382: Is Chaos^2NLI a widely used subset and shorthand? If so, cite it. If not, I'd suggest changing the shorthand, as the Chaos^2NLI ends up looking like ChaosNLI with a footnote, especially since the "squared" part doesn't have clear semantics.

Figure 3: The label shorthand was not apparent on first pass and required scrolling back and forth between table 1 and figure 3 to figure out the mapping. I'd try to establish this shorthand earlier in the work so that readers expect it throughout rather than changing shorthand depending on the space limitations of different tables.

Figure 5: It's not clear what this figure is communicating. What is the ideal trend we'd like to have in this graph? A linear relationship between entropies? If so, communicate that in either text or visualization. How do we expect the trends to change when delineated by categories? Are we hoping to see a more clear trend for some categories than for others? If so, communicate this in the text. Right now, these figures take up a good bit of space, but they just seem to communicate the same info as Figure 4 in a different format.

---

> ### Author Rebuttal · Authors · 2023-08-29
>
> Dear Reviewer DgQC,
>
> Thank you for your appreciation of our work and insightful review. We are grateful that you find our problem, which is modeling human disagreement with LLMs, crucial. We are also encouraged that you see the potential of our proposed metrics to have impacts in LLM and human opinion distribution domains.
>
> Below, we have tried best to address your questions and concerns. Please take a look at our response and let us know if further clarification is needed.
>
> **Q1) You cite the decoding strategy as a limiting factor for exploring the whole space. How does this apply to your NLI tasks, where the model is selecting between a finite number of options rather than performing open-ended decoding? With lines 162-163, it seems like you are saying each class label will be inferred based on the open-ended generation, but this isn't totally clear.’**
>
> As mentioned in lines 154-156, using decoding strategies such as beam search might hinder models to explore the entire space. Within the application to our NLI task, the decoding strategy is not a major problem for LPE which requires a single output token. However, for MCE, which we do not limit the number of tokens, we are able to mitigate this source of limitation as the LLM is able to output a wider set of answer space than a single generation equipped with a decoding strategy can.
> Also, you are correct on “*each class label will be inferred based on the open-ended generation.*” We will clarify this in more detail: we prompt instruction fine-tuning LLMs to select between a finite number of options (ex., entailment, contradiction, and neutral for ChaosNLI-M/S datasets) and to perform open-ended decoding, where the models have to predict the entire string approximating valid options. Please look into our Prompt examples in Appendix (p. 14) to see the exact prompt we used for some instances.
>
> **Q2) Line 182 - 185, suggest the probability mass is non-constant, but this is a choice by this calculation. Why don't you use the adjustment from Santurkar et al. 2023 in your citations to make the probability mass constant by estimating the mass of missing options as their upper bound?**
>
> The unknown probability mass is displayed due to two reasons: a missing option or the model's characteristic to be unconfident. Santurkar et al. assigns the probability mass of missing options to the minimum value of the remaining mass or the minimum probability assigned to any of K token choices (p. 29 in Santurkar et al. 2023) and normalizes after, which will lead to an unsound overestimation of unspecified (i.e., missing) options. We make adjustments by not assigning probability mass for the missing options and directly normalize our output which might be overestimating the probabilities of existing options, but refraining from any problematic assignment to missing options.
>
> **Q3) The FLAN Models use the full ANLI and QNLI training datasets as part of their instruction fine tuning. Why are they marked as Partial Exposure?**
>
> Thank you for pointing this out. Here we provide more details to clarify the concept of Partial Exposure (PE). Our intent in marking FLAN models as PE is because they are not directly fine-tuned with only NLI datasets. In fact, yes, it is true that they are fine-tuned with ANLI and QNLI datasets, however, in the FLAN paper (Chung et al., 2022), it is stated that “*We apply the maximum cap for each task because there are tasks that are much larger than others in the same mixture, which can dominate the sampling.*” This suggests that they also have used tasks other than NLI to fine-tune the FLAN models.
>
> In addition, in Table 23 (p. 46 in Chung et al., 2022), the maximum cap and proportion rates for the Muffin mixture, which contains 80 tasks including the NLI tasks, are capped with at most 30k instances per task. This is a smaller proportion compared to the entire training dataset size of the full ANLI (r3: 100k,  r2: 45.5k, and r1: 16.9k, thus, theoretically possible to be fully sample r1, but highly unlikely) and QNLI (116k) datasets. To get a better sense of the utilized NLI proportion, in Table 22 (p. 46 in Chung et al., 2022), FLAN-T5-XXL is fine-tuned with a batch size of 64 for 14K steps, summing up to approximately 896K instances, and the FLAN models utilize a mixture weighting around 40-50% for the Muffin task. Hence, we can conclude that unlike the Full Exposure (FE) models which are directly fine-tuned to only NLI tasks, the FLAN models are assigned as PE models due to comparably under usage of NLI datasets for finetuning, according to our definition. We will clarify the definition of PE in Section 4.2 in the later version of this paper.
>
> **Q4) How can you confidently say that Davinci 2 has no exposure to these tasks?**
>
> We were also aware of the uncertainty of whether Davinci 2 has exposure to NLI tasks or not. Ouyang et al. elaborate that “*the dataset consists primarily of text prompts submitted to the OpenAI API … Playground interface*,” thus; unless the authors added a small proportion of NLI data or the API users input NLI problems out of their own interests, the model would not have been fine-tuned to the NLI dataset.
>
> However, to be sure, we reached out to one of the authors of the paper and received confirmation that no NLI dataset and/or instances were used in their first response. Unfortunately, another author responded later that they, in fact, do have NLI instances but with a number less than 10. We will add a Minimal Exposure (ME) level to have Davinci models separated from Partial Exposure (PE) FLAN models with thousands of NLI supervision instances and with No Exposure (NE) model to encourage a fairer comparison.
>
> **Q5) Why can't you compare the probabilities of the first token of each option for OS LPE?**
>
> We try to respond to this issue in the paper by lines 264-268, *“OS, on the other hand, is not considered in the LPE formulation for simplicity in both implementation and application, where the token-specific probability of a varying length and output format has to be mapped per class.”*
>
> To further elaborate, our main purpose of the paper as you kindly phrase is to *develop a sound metric of mapping LLMs to distributions of opinions has potential impacts in each of those domains*. If we allow LPE-OS, we would have to allow multiple tokens to be generated which will lead to an exhaustive, rule-based processing: 1) saving log probabilities of every token, 2) selecting the determining token probabilities that may vary by instance/dataset/task, and 3) mapping the designated probability to the correct class label. For example, FLAN models’ frequent representation of the label “neutral” was by an answer “It is not possible to tell”. To incorporate the generation characteristics for every used model along with the varying deterministic token position, we thought it best to leave out LPE–OS to encourage a scalable, comprehensive generation/evaluation strategy to estimate human disagreement distribution.
>
> **The fundamental comparison seems to incompletely answer the question layed out in the title, confounded by the way the models are compared to human annotators. The probability distribution of labels assigned by different human annotators is the average of opinions from multiple agents. However, disagreement does not neccesarily mean that each individual annotator was uncertain in the label they selected. Given the high variance from prompting, it seems not sufficiently sound to compare "agreement" of LLMs by using the sampled variance over one prompt.**
>
> Yes, we appreciate and understand your concerns that using sampled responses over one prompt may not perfectly align with human disagreement distribution. Our title, “Can LLMs Infer and Disagree Like Humans?” may lead to confusion that we use multiple LLMs to estimate the human disagreement distribution. However, our intention with the title is whether each of the state-of-the-art instruction fine-tuning LLMs, with their emergent ability, are capable of representing the dissenting voices of humans. Since LLM decodes different responses even when prompted with the same question, we explore whether utilizing varying responses over n instances (MCE) or top-k tokens over a single instance (LPE) might be enough to represent the human disagreements.
>
> We believe your concern on “*disagreement does not necessarily mean that each individual annotator was uncertain in the label they selected*,” especially roots in our LPE method since it is calculated with a single instance. Although we regard MCE as a more intuitive, human-like way to represent disagreement (e.g., aggregating voices of n=100 people), in Table 2, we can observe how LPE outperforms MCE in several models like GPT-3.5-D3 and Stable Vicuna with higher efficiency (i.e., less cost for the number of instances).
>
> We also want to point out that we do not use the sampled “variance” to calculate uncertainty. Rather, we aggregate the LLM uncertainty over one prompt using either the LPE or MCE method to estimate human disagreement distribution. While Zhou et al. injected expressions of uncertainty into prompts (e.g., “I think the answer is …”) to increase the accuracy in terms of predicting one ground-truth answer correctly by generating 10 tokens (p. 4 in Zhou et al.), our work not only aims to achieve high accuracy but also to capture different perspectives of humans with MCE/LPE.
>
> As you mention, high prompt sensitivity can possibly debunk the soundness of our paper. We try to mitigate this problem by shuffling the order of the class options and averaging them over three trials. We are aware of the effect of the variability occurring in answers generated with different instruction prompts. However, Chiang et al. claim that variants in prompts do not affect the overall conclusion, suggesting that averaging different outputs generated from slightly different prompts with MCE/LPE may be enough to draw the general conclusion that each generative LLM cannot infer like humans and represent their disagreements.
>
> **The paper does not clearly communicate actionable insights from their work. This work has the potential to more systematically understand whether we can map model uncertainty to human uncertainty, but at present it misses that mark.**
>
> We agree that the present work does not clearly address actionable insights, hence, we extend the discussion below and will rephrase the introduction and discussion sections to better communicate: Compared to previous works you mentioned, such as Santurkar et al. and Gillardi et al., our work is the first study to explore whether billion-scale generative LLMs could capture human disagreements in the most fundamental NLP task, NLI. Santurkar et al. found “misalignment between the views of current LLMs and US demographic groups,” by prompting personal sensitive questions. One of the reasons for this misalignment issue is stated as “the left-leaning tendencies in human feedback” in the paper, which can be easily inferred from the result of current LLM steering towards particular demographic groups.
>
> However, it is more difficult to find the explicit reason for misalignment between current LLMs and humans on NLI tasks since there might be diverse sets of reasons leading to human disagreement, such as annotator disagreement, subjectivity, and multiple plausible answers (Plank et al., 2023). We try to find a subtler insight into why the LLMs disagree in some instances by a simple regression model (Figure 5) on LLM vs. human disagreement level (i.e., entropy) per instance. However, we did not find a direct relationship between the reasons of humans and LLM entropy level (See “What causes LLM to disagree?” in Discussion). In conclusion, we allude to the need to reflect a larger population in generative LLMs into less-sensitive issues by our experimentation on NLI, a simple yet effective testbed to represent casual disagreements.
>
> We also deeply appreciate you for the comments on missing references and typos/presentation improvements to bring a stronger presentation for the paper. We will respond to some of them:
>
> **Missing References:**
>
> We will integrate the following important references in our Related Works section to clarify and strengthen our position:
>
> •	Intercoder agreement results (Gilardi et al., 2023)
>
> •	Constrained decoding & logit bias adjustments to LLMs to generate a more confident distribution output. (Ziems et al., 2023)
>
> •	Explicit expression of uncertainty can be exploited to better performance (Zhou et al., 2023)
>
> •	The limitation of status-quo model training which leads to poor calibration towards human perspectives (Gordon et al., 2021)
>
>
> **Typos Grammar Style And Presentation Improvements:**
>
>
> •	**[GPT Naming Convention]** Thank you for this suggestion. We will change GPT-3.5-D2/D3 to GPT-3-D2/D3.
>
>
> •	**[Chaos^2NLI]** We will change Chaos^NLI to HighChaosNLI to avoid confusion on the “squared” part.
>
>
> •	**[Figure 3 Label Shorthand]** Thank you for this suggestion. We will put the same shorthand of model names across all the figures and tables (e.g., Flan-T5-XXL to FlanXXL).
>
>
> •	**[Figure 5 Better Communication]** Yes, it would be ideal to have a linear relationship between humans and LLM entropies. We will add the y=x line to show the ideal case. We were expecting to see whether there is a relationship between human disagreement reason categories and LLM entropy level. If there is, we would have observed the same colored dots clustered with similar y-axis values. However, this was not the case. We also want to point out that whereas Figure 4 presents the overall counts of entropy levels for humans and each model, Figure 5 shows the relationship between human and model entropy levels for each instance, and the instance is categorized by human disagreement sources. Thank you for your suggestion on improving the clarity of uncertain parts in the paper. We will surely extend this explanation in or after lines 422-427.
>
> **References:**
>
>
> •	Santurkar, S. et al. (2023). Whose Opinions Do Language Models Reflect? In International Conference on Machine Learning, pages 29971-30004.
>
> •	Chung, H. W. et al. (2022). Scaling instruction-finetuned language models. arXiv preprint arXiv:2210.11416.
>
> •	Ouyang, L. et al. (2022). Training language models to follow instructions with human feedback. In Advances in Neural Information Processing Systems, Volume 35, pages 27730-27744.
>
>
> •	Zhou, K., Jurafsky, D., & Hashimoto, T. (2023). Navigating the Grey Area: Expressions of Overconfidence and Uncertainty in Language Models. arXiv e-prints, arXiv-2302.
>
>
> •	Chiang, C. H., & Lee, H. Y. (2023). Can Large Language Models Be an Alternative to Human Evaluations?. In Proceedings of the 61st Annual Meeting of the Association for Computational Linguistics, Volume 1: Long Papers, pages 15607–15631.
>
>
> •	Gilardi, F., Alizadeh, M., & Kubli, M. (2023). Chatgpt outperforms crowd-workers for text-annotation tasks. In Proceedings of the National Academy of Sciences of the United States of America, Volume 120, No 30, pages 1-3.
>
>
> •	Plank, B. et al., (2022). The “problem” of human label variation: On ground truth in data, modeling and evaluation. In Proceedings of the 2022 Conference on Empirical Methods in Natural Language Processing, pages 10671–10682.

---

### Official Review · Reviewer_qn54 · 2023-08-05

**Typos Grammar Style And Presentation Improvements:** 1. I would have liked to see another …
**Soundness:** 3

**Excitement:**

4: Strong: This paper deepens the understanding of some phenomenon or lowers the barriers to an existing research direction.

**Paper Topic And Main Contributions:**

This paper focuses on the NLI task and studies the uncertainty in the learned distribution of LLMs and how it differs from human disagreement. The paper also proposes two simple techniques to estimate LLM's learned distribution to overcome the challenge of API-based LLMs that limits the number of queries you can make.



**Questions For The Authors:**

1. Both MCE and LPE rely on a set of valid options for each class. How many valid options do you have per class, and do you have an idea how sensitive the estimated distribution is to the choice of these valid options?
2. How slow/fast are MCE and LPE? NLI is a relatively simple task compared to others with long outputs, so it'd be interested to know scalable they are to more complicated tasks.
3. In section 5, the first conclusion is about how model size alone isn't a critical factor for high performance on NLI. But based on my reading of Table 1, it's really GPT-3.5 that's clearly the outlier. I feel like it says more about GPT-3.5's unique properties rather than its size -- within the same model family, you do see model size matter (see various sizes of Flan models, and OPT-IML models).
4. I might have missed this, but what exactly do you mean by 'shuffled option selection'?
5. I'm not sure I understand the argument in line 393-397 -- what's the characteristic of the dataset that led you to conclude it's mere coincidence?

**Reasons To Accept:**

1. Efficiently quantifying the degree to which LLM's distribution differs from human agreement/disagreement is a valuable effort, esp given the challenge of API-based models that are becoming more and more in use nowadays.
2. Very clearly written, and very thorough set of experiments that make the main point of the paper that LLMs fail to adequately capture human disagreement.

**Reasons To Reject:**

My main concern is regarding how applicable MCE and LPE methods are for other more complicated tasks. See the Questions section for more details.

**Reproducibility:**

4: Could mostly reproduce the results, but there may be some variation because of sample variance or minor variations in their interpretation of the protocol or method.

**Reviewer Confidence:**

3: Pretty sure, but there's a chance I missed something. Although I have a good feel for this area in general, I did not carefully check the paper's details, e.g., the math, experimental design, or novelty.

---

> ### Author Rebuttal · Authors · 2023-08-29
>
> Dear Reviewer qn54,
>
> Thank you for your thoughtful review. We are glad that you find our attempt to quantify and analyze the alignment between the LLM distribution and human disagreement distribution using various LLMs to be thoroughly conducted and valuable.
>
> Below, we have tried our best to address your questions and concerns. Please take a look at our response and let us know if further clarification is needed.
>
> **Q1) Both MCE and LPE rely on a set of valid options for each class. How many valid options do you have per class, and do you have an idea how sensitive the estimated distribution is to the choice of these valid options?**
>
> | | | *ChaosNLI* | *$\alpha$* | *ChaosNLI* | *S* | *ChaosNLI* | *M* |
> |------------------|:-----:|:----------:|:-------:|:----------:|:-----:|:----------:|:-----:|
> | | | *MCE* | *LPE* | *MCE* | *LPE* | *MCE* | *LPE* |
> | *FLAN-T5-XXL* | *OS* | 99.96% | - | 89.24% | - | 90.85% | - |
> | *FLAN-T5-XXL* | *NS* | 99.74% | 99.81% | 99.49% | 99.45% | 99.51% | 93.55% |
> | *GPT-3.5-D3* | *OS* | 100.00% | - | 99.91% | - | 99.99% | - |
> | *GPT-3.5-D3* | *NS* | 99.44% | 99.57% | 99.93% | 99.74% | 99.83% | 99.15% |
> | *Stable Vicuna* | *OS* | 97.84% | - | 85.20% | - | 86.04% | - |
> | *Stable Vicuna* | *NS* | 99.57% | 99.53% | 99.28% | 99.39% | 99.30% | 99.36% |
>
> Table A | The Percentage of Valid Options for Three Models - Flan-T5-XXL, GPT-3.5-D3, and Stable Vicuna on Subsets of ChaosNLI Datasets. We present the average results of ChaosNLI-α, S, and M; for each dataset, we randomly sample 100 instances. For MCE, we record the percentage of the number of captured valid options out of a total of 100 generated outputs. For LPE, we report the sum of the probability of valid options captured within the top-k (k=5) tokens.
>
> Our implementation of the parsing of answers relies on string matching the stems of the candidate space that we sample out in the initial stage of experiments rather than having all the variations of each class label counted. As a matter of fact, the distribution was not sensitive to these options as most models were capable of generating a combination of strings that was parsable into a single class label for most (> 99%) time (Table A). Furthermore, by prompting the model to generate a number (Number Selection; NS) rather than the actual option (Option Selection; OS), the models were able to perfectly parse out the generated option into a specific class.
>
> **Q2) How slow/fast are MCE and LPE? NLI is a relatively simple task compared to others with long outputs, so it'd be interested to know scalable they are to more complicated tasks.**
>
> | | | *ChaosNLI* | *$\alpha$* | *ChaosNLI* | *S* | *ChaosNLI* | *M* |
> |------------------|:-----:|:----------:|:-------:|:----------:|:-----:|:----------:|:-----:|
> | | | *MCE* | *LPE* | *MCE* | *LPE* | *MCE* | *LPE* |
> | *FLAN-T5-XXL* | *OS* | 8.21 | - | 8.28 | - | 8.94 | - |
> | *FLAN-T5-XXL* | *NS* | 3.91 | 0.16 | 3.14 | 0.15 | 3.49 | 0.16 |
> | *GPT-3.5-D3* | *OS* | 4.93 | - | 3.92 | - | 4.11 | - |
> | *GPT-3.5-D3* | *NS* | 3.55 | 0.37 | 3.47 | 0.39 | 3.57 | 0.37 |
> | *Stable Vicuna* | *OS* | 13.97 | - | 10.58 | - | 11.61 | - |
> | *Stable Vicuna* | *NS* | 5.64 | 0.12 | 3.94 | 0.09 | 4.48 | 0.09 |
>
> Table B | Comparison of Inference Time Between MCE and LPE for Models - Flan-T5-XXL, GPT-3.5-D3, and Stable Vicuna on Subsets of ChaosNLI Datasets. We present the average results of ChaosNLI-$\alpha$, S, and M; for each dataset, we randomly sample 100 instances. For both MCE and LPE, we record the generation time in seconds using a Python package called “microbench.”
>
> Referring to lines 156 (for MCE) and 177-178 (for LPE) of the paper, LPE is n times faster than MCE since LPE requires only one sample to be generated (O(1)), whereas MCE must produce n samples (O(n)) to estimate the final distribution capturing disagreement. We further provide an empirical comparison of the average inference time per instance between MCE and LPE (Table B).
>
> **Q3) In section 5, the first conclusion is about how model size alone isn't a critical factor for high performance on NLI. But based on my reading of Table 1, it's really GPT-3.5 that's clearly the outlier. Within the same model family, you do see model size matter.**
>
> You are on point regarding that size is a very critical component for the NLI performance within the same model family. However, as mentioned in lines 282-283 and 307-310 of the paper, we want to emphasize that size alone is not the sole factor that can affect NLI performance. In fact, the NLI exposure seems to be a more important factor since a 355 million parameter RoBERTa-Large can significantly outperform a 175 billion model GPT-3.5-D2/D3. Still, we deeply understand your comment on our expression which can be better rephrased in a less misleading way. We will revise the first Results section based on your valuable suggestion.
>
> **Q4) Explain Shuffled Option Selection.**
>
> We will clarify that as the details of the shuffling are laid out in Appendix A, but not in the main paper. The shuffled option selection basically indicates randomly permuting the class options (e.g., 1, 2, 3 to 3, 1, 2). This is needed to reduce the variability of performances caused by the high prompt sensitivity of generative LLMs (Hendrycks et al., 2021)  to increase the robustness of performances (Santurkar et al., 2023). In fact, the prompt sensitivity of LLMs also resembles the sensitivity of the position incurred in human respondents (Santurkar et al., 2023), which is why we thought it more natural to adopt this method.
>
> **Q5) What's the characteristic of the dataset that led you to conclude it's mere coincidence?**
>
> As you can observe in Table 1 and Figure 3, Stable Vicuna is certainly not the best model in terms of accuracy (NLI Performance) and JSD/DCE (human alignment performance) for the full ChaosNLI datasets. However, when we test it on top-100 human disagreeing instances (which we entitle, Chaos^2NLI), Stable Vicuna “seemingly” becomes the best model in human alignment with the lowest JSD/DCE among all the other models (Table 3).
>
> This phenomenon is not due to the fact that Stable Vicuna could estimate the most disagreeable instances well, but because *Stable Vicuna is uncertain in most instances since it displays higher entropy levels compared to the other models (Figure 4).* Hence, paired with the hint of the worst accuracy out of all the models for full ChaosNLI datasets but with the best human alignment performance score for top-100 human disagreeing instances, we conclude that it is a mere coincidence that Stable Vicuna exhibits the best performance in terms of human alignment performances in Chaos^NLI dataset. We will clarify this aspect more in detail in the Results section.
>
> **My main concern is regarding how applicable MCE and LPE methods are for other more complicated tasks**
>
> We understand your concern about how MCE and LPE methods can be applied to further complicated tasks other than NLI. We briefly mention the possibility of transferring other NLP tasks to the NLI task (lines 488-505). In addition to this, if these complicated tasks can be formulated into multiple-choice style formats, with the assumption that our models can generate valid options for each class, our methods are easily scalable as Table B from Q2) suggests. For example, more complex tasks such as ambiguous fact verification (Glockner et al., 2023) or TruthfulQA (Lin et al., 2022) are in or are capable of directly being formulated into a multiple choice format; thus, our methods are easily applicable. However, with tasks such as Abstractive QA such as HotpotQA (Yang et al., 2018) that can’t be directly transferred, we can exploit the power of LLMs to perform semantic matching (Kamalloo et al., 2023) and even further.
>
> However, it should be noted how our methods are only useful in tasks where “human disagreement” is a significant signal that needs to be captured. For example, since it is important to include major and minor opinions, we can easily apply our methods to detect disagreements in hate speech (Schmidt et al., 2017). In contrast, spotting disagreement in the arithmetic reasoning task (Cobbe et al., 2021) might be not as important since it often requires a logical step-by-step reasoning procedure to obtain an accurate answer.
>
> We also deeply appreciate you for the comments on missing references and typos/presentation improvements to bring a stronger presentation for the paper. We will respond to some of them:
>
> **Typos Grammar Style And Presentation Improvements:**
>
>
> •	**[Table 1 Revision]** Thank you for this suggestion. We will add the model parameters in Table 1.
>
>
> •	**[Figure 3 Y-axis Label Revision]** Thank you so much for pointing this out. We will revise the label as you suggested.
>
> **References:**
>
> •	Hendrycks, D. et al., (2021). Measuring Mathematical Problem Solving With the MATH Dataset. Sort, 2(4), pages 0-6.
>
>
> •	Santurkar, S. et al. (2023). Whose Opinions Do Language Models Reflect? In International Conference on Machine Learning, pages 29971-30004.
>
>
> •	Glockner, M. et al., (2021). AmbiFC: Fact-Checking Ambiguous Claims with Evidence. arXiv e-prints, arXiv-2104.
>
>
> •	Lin, S. et al., (2022). TruthfulQA: Measuring How Models Mimic Human Falsehoods. In Proceedings of the 60th Annual Meeting of the Association for Computational Linguistics, pages 3214-3252.
>
> •	Yang, Z. et al. (2018). HotpotQA: A Dataset for Diverse, Explainable Multi-hop Question Answering. In Proceedings of the 2018 Conference on Empirical Methods in Natural Language Processing, pages 2369–2380.
>
>
> •	Kamalloo, E., Dziri, N., Clarke, C. L., & Rafiei, D. (2023). Evaluating Open-Domain Question Answering in the Era of Large Language Models. In Association for Computational Linguistics, pages 5591–5606.
>
> •	Cobbe, K. et al. (2021). Training verifiers to solve math word problems. arXiv preprint arXiv:2110.14168.
>
> •	Schmidt, A., & Wiegand, M. (2017). A survey on hate speech detection using natural language processing. In Proceedings of the fifth international workshop on natural language processing for social media, pages 1-10.

---

### Meta-Review · Area_Chair_gYGy · 2023-09-19

**Recommendation:** 4

**Metareview:**

This paper presents examines alignment of LLMs with humans on the task of natural language inference. Thorough experiments on many models indicate that LLMs fail to adequately capture human disagreement. Concerns about the paper include the lack of actionable insights and the framing of the paper, which were addressed in the lengthy rebuttals.

---

### Decision · Program_Chairs · 2023-10-07

**Decision:**

Accept-Main

**Comment:**

This paper presents examines alignment of LLMs with humans on the task of natural language inference. Thorough experiments on many models indicate that LLMs fail to adequately capture human disagreement. Concerns about the paper include the lack of actionable insights and the framing of the paper, which were addressed in the lengthy rebuttals.